# Mechanisms of triplet energy transfer across the inorganic nanocrystal/organic molecule interface

Xiao Luo [1], Yaoyao Han[1,2], Zongwei Chen[1], Yulu Li[1], Guijie Liang[3], Xue Liu[1], Tao Ding[1], Chengming Nie[4], Mei Wang[4], Felix N. Castellano [5] & Kaifeng Wu[1]*

The mechanisms of triplet energy transfer across the inorganic nanocrystal/organic molecule interface remain poorly understood. Many seemingly contradictory results have been reported, mainly because of the complicated trap states characteristic of inorganic semi-conductors and the ill-defined relative energetics between semiconductors and molecules used in these studies. Here we clarify the transfer mechanisms by performing combined transient absorption and photoluminescence measurements, both with sub-picosecond time resolution, on model systems comprising lead halide perovskite nanocrystals with very low surface trap densities as the triplet donor and polyacenes which either favour or prohibit charge transfer as the triplet acceptors. Hole transfer from nanocrystals to tetracene is energetically favoured, and hence triplet transfer proceeds via a charge separated state. In contrast, charge transfer to naphthalene is energetically unfavourable and spectroscopy shows direct triplet transfer from nanocrystals to naphthalene; nonetheless, this "direct" process could also be mediated by a high-energy, virtual charge-transfer state.

[1] State Key Laboratory of Molecular Reaction Dynamics and Dynamics Research Center for Energy and Environmental Materials, Dalian Institute of Chemical Physics, Chinese Academy of Sciences, Dalian, Liaoning 116023, China. [2] University of the Chinese Academy of Sciences, Beijing 100049, China. [3] Hubei Key Laboratory of Low Dimensional Optoelectronic Materials and Devices, Hubei University of Arts and Science, Xiangyang, Hubei 441053, China. [4] State Key Laboratory of Fine Chemicals, Institute of Artificial Photosynthesis, Dalian University of Technology, Dalian, Liaoning 116024, China. [5] Department of Chemistry, North Carolina State University, Raleigh, NC 27695-8204, USA. *email: kwu@dicp.ac.cn

Molecular spin-triplet states (hereafter called "triplets") have been an important subject of research in photochemistry for many decades[1]. The long lifetime of these triplets allows for the time window for a variety of chemical reactions, enabling their applications in photoredox catalysis[2,3]. They are also engaged in the generation of singlet oxygen which is essential for type II photodynamic therapy[4]. Moreover, molecular triplets have recently become a fascinating topic in the field of solar energy because of the phenomena of singlet fission (SF)[5], whereby two triplets are generated by the fission of one singlet excited state, and an essentially opposite process called triplet fusion or triplet–triplet annihilation that can be used for photon upconversion (TTA-UC)[6]. SF and TTA-UC can alleviate the thermalization and transmission losses, respectively, in traditional solar conversion schemes, holding promise for breaking the Shockley-Queisser limit for the efficiency of single-junction solar cells.

Recently, there has been growing interest in combining the light harvesting and emitting capability of inorganic semiconductors with the SF and TTA-UC processes in organic molecules via interfacial excitation energy transfer to explore new functionalities[7–11]. For example, researchers have pursued the ideas of harvesting the SF-induced triplets of polycyclic aromatic hydrocarbons (PAHs) such as tetracene and pentacene using either silicon solar cells[12,13] or lead chalcogenide nanocrystals (NCs)[7,8,14] for efficiency-doubled light conversion or emission. Alternatively, inorganic semiconductors, and in particular their NCs, can be used to sensitize the triplets of PAHs for TTA-UC[9,15–19]. The advantage of these inorganic sensitizers over traditional molecular sensitizers lies in the much weaker bright-dark states splitting (a few to 10 s of meV) in inorganic semiconductors compared with molecules (100 s of meV) allowing for a higher upconversion energy gain[20,21]. In addition to unidirectional harvesting or sensitization of molecular triplets using semiconductors, reversible energy cycling between them has also been observed[22], enabling new phenomena including thermally activated delayed emission from semiconductor NCs[23,24].

A key process underlying the above mentioned functionalities is triplet energy transfer (TET) between inorganic semiconductors and organic molecules. TET is in general believed to proceed via the short-range, Dexter-type mechanism[25]. Therefore, TET is an interfacial process and semiconductor NCs are well suited for studying these processes as diffusion processes inside the bulk can be neglected. Previous studies on TET from NCs to PAHs, however, have resulted in contradictory models. For CdSe NC-anthracene or pyrene systems, a direct Dexter TET mechanism was proposed on the basis of the correlated CdSe signal decay and molecular triplet formation and the absence of molecular cation radicals[10]. For PbS NC-tetracene or pentacene systems, at least three types of models have been proposed, including hole transfer mediated TET[26], surface states mediated TET[27] and direct TET competing with hole transfer[28]. Thus, a unified picture for the mechanism of TET across the inorganic/organic interface is still lacking.

Here we study TET mechanisms by using lead halide perovskite NCs as the triplet donor and tetracene and naphthalene as the molecular triplet acceptors which favour and prohibit hole transfer from NCs, respectively. The high emission quantum yields (QYs) of the CsPbBr₃ NCs exclude a major impact from surface states. We report definitive spectroscopic evidence for hole transfer mediated TET in the case of NC-tetracene. As for NC-naphthalene, we observe direct formation of molecular triplets, but a coupling matrix element analysis indicates that TET could also be mediated by a virtual charge-transfer (CT) state.

## Results

**System design and energetics analysis.** We use CsPbBr₃ perovskite NCs as the triplet donor because of their "defect-tolerance" enabling high photoluminescence (PL) QYs (~50–90%), which have already found promising applications for light-harvesting and -emitting devices[29–31]. In contrast, traditional CdSe or PbS NCs typically possess much lower QYs (< 20%) if not specially engineered, making the interfacial TET process inevitably complicated by the presence of surface trap states. Moreover, as will be elaborated below, the co-contributions of the electron and the hole to the transient absorption (TA) features of CsPbBr₃ NCs, in conjunction with their strong PL, allow us to perform a combined TA and time-resolved PL study to unambiguously establish the role of CT states in mediating TET. In our study, we mainly used a sample of quantum-confined CsPbBr₃ NCs with an average diameter of $3.8 \pm 0.4$ nm (Fig. 1a and Supplementary Fig. 1), but NCs of other sizes have also been studied for control experiments; details regarding sample preparation and characterization are provided in the Methods section. Our recent studies have shown that quantum confinement is essential for efficient TET from CsPbBr₃ NCs as it provides the electronic coupling required for Dexter-like TET[32,33].

We used two carboxyl-functionalized PAH molecules, 1-naphthalene carboxylic acid (NCA) and 5-tetracene carboxylic acid (TCA), as the triplet acceptors (Fig. 1b). The redox potentials of unsubstituted naphthalene and tetracene have been reported[1], but, considering the potential effect of the carboxyl group, we measured the redox potentials of NCA and TCA molecules using cyclic voltammetry (CV); see Methods for details. For both NCA and TCA, only their oxidation potential energies ($E_{ox}$) could be determined from CV in the used electrochemical window (Supplementary Figs. 2, 3) and their reduction potential energies ($E_{red}$) were approximated as[3,34,35]: $E_{red} = E_{ox} + E_g$, with $E_g$ being the optical gaps of the molecules. Similarly, the reduction potential energies to form molecular triplets ($E_{red,T}$) were approximated as[3,34,35]: $E_{red,T} = E_{ox} + E_T$, with $E_T$ being the triplet energies. For CsPbBr₃ NCs, both the lowest electron and hole energies ($E_e$ and $E_h$, respectively) were determined from CV (Supplementary Fig. 4), with some additional intragap features that were tentatively assigned to lead oleate species in the solution[36].

The determined energy level alignments in the NC-PAH systems are summarized in Fig. 1c. The energy levels of NCA and TCA are indeed shifted by 100 s of meV compared with those reported for unsubstituted naphthalene and tetracene[1]. The energy difference between $E_e$ and $E_h$ determined for the NCs is ~2.9 eV, which is consistent with the optical gap of 2.7 eV (460 nm) after accounting for an electron-hole binding energy of ~0.2 eV in 3.8-nm CsPbBr₃ NCs (Supplementary Note 1). This consistency also suggests that while the absolute values of the energy levels in Fig. 1c are only for reference due to many well-known complications associated with these measurements, the energy level alignments (i.e., relative values) should be reliable.

The driving forces for CT reactions ($-\Delta G_{CT}$), however, are not simply determined by the "single-particle" energy alignments shown in Fig. 1c. Rather, we need to account for various Coulombic binding and charging energies involved in CT. For example, when examining hole transfer from NCs to NCA, we should consider the energy penalty associated with breaking the electron-hole pair in NCs and putting extra charges into NCs and NCA as well as the energy compensation from electron-hole binding in the charge separated states (NC⁻-NCA⁺). Detailed for these calculations are provided in Supplementary Note 1. In Fig. 2, we plot the calculated energies of various CT states (NC⁻-NCA⁺, NC⁺-NCA⁻, NC⁻-TCA⁺ and NC⁺-TCA⁻) relative to that of photoexcited NCs (NC*) in a Marcus-type reaction coordinate diagram; see also Supplementary Table 1. According to the

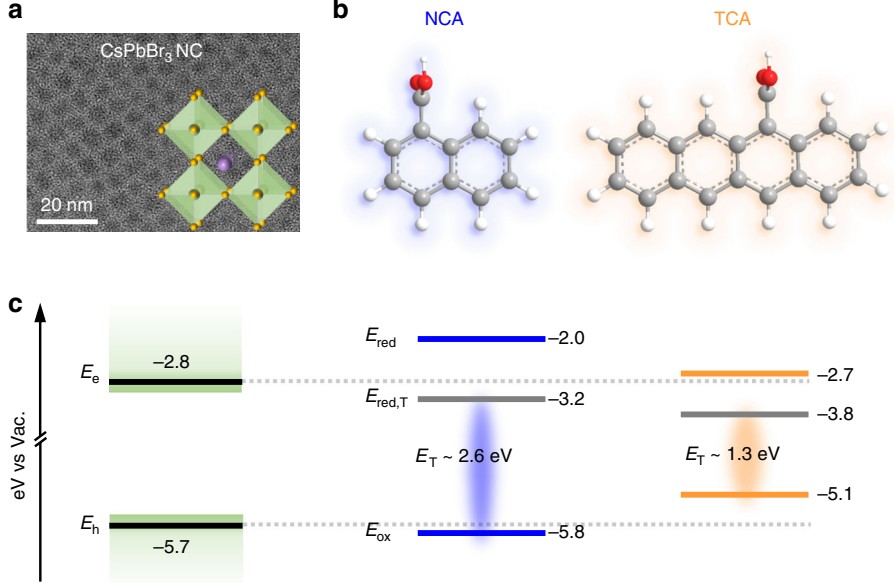

**Fig. 1 Design of the NC-PAH systems for TET study. a** A representative transmission electron microscopy (TEM) image of the CsPbBr$_3$ NCs. The scale bar is 20 nm. Inset: a schematic structure of the perovskite lattice. **b** Molecular structures of the 1-naphthalene carboxylic acid (NCA) and 5-tetracene carboxylic acid (TCA). **c** Schematic energy level alignment between NCs and NCA and TCA determined from cyclic voltammogram. $E_e$ and $E_h$ are the lowest electron and hole energy levels in the conduction and valence bands, respectively. The difference between them is ~2.9 eV, which is higher than the optical gap of the NCs (~2.7 eV) because of a strong electron-hole coulomb binding energy in NCs. $E_{ox}$, $E_{red}$ and $E_{red,T}$ are the ground-state oxidation potential energy, reduction potential energy and triplet state reduction potential energy, respectively, of the PAH molecules. $E_T$ is the triplet energy.

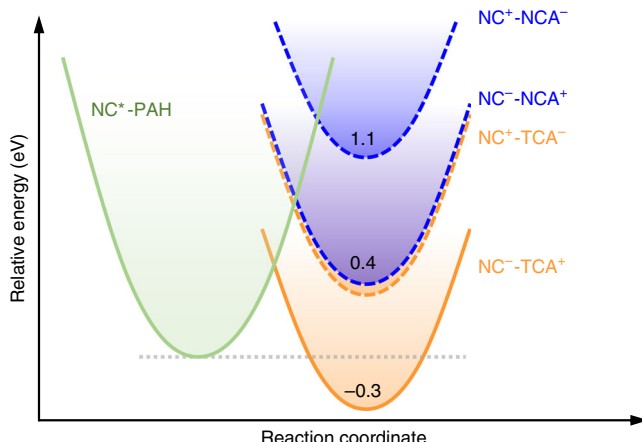

**Fig. 2 CT energetics.** The parabolas represent the reactant state (NC*-PAH) and the various CT product states drawn on a continuous reaction coordinate mainly contributed by the surrounding medium. The lowest energies of the CT states with respect to that of the NC*-PAH state are indicated. Note that the curvatures of these parabolas and the horizontal displacements between them are not for quantitative use, because the reorganization energies for NCA and TCA molecules should be different.

diagram, electron transfer from photoexcited NCs to both NCA and TCA is energetically unfavourable. On the other hand, hole transfer from photoexcited NCs to ground-state NCA and TCA should be energetically disallowed and favoured, respectively.

TET from photoexcited NCs to NCA and TCA is energetically allowed in both cases because the triplet energies of NCA (~2.6 eV)[3] and TCA (~1.3 eV)[37] are lower than that of the NC band edge exciton (~2.7 eV), but may proceed via different mechanisms on the basis of their different CT possibilities.

**Optical properties**. NCA and TCA molecules were anchored onto NC surfaces via their carboxyl groups by replacing part of the native ligands on NCs[10,23]; see Methods for details. The final samples were dispersed in hexane in which NCA and TCA have negligible solubility to ensure that the PAH molecules remaining in the hexane solution were mostly bound to NC surfaces. The absorption and PL spectra of free NCs are displayed in Fig. 3a. The lowest energy absorption peak of CsPbBr$_3$ NCs is situated at ~460 nm, which is blue-shifted from CsPbBr$_3$ bulk (~520 nm) due to the quantum confinement effect. The PL peak is ~474 nm, corresponding to a Stokes shift of 80 meV. The symmetric PL band and the absence of a low-energy, trap-related emission band are consistent the high PL QY (~70%) of the sample. Figure 3b displays the absorption spectra of the NC-PAH complexes, which shows additional absorptive features from the NCA and TCA molecules in addition to NC absorption. By performing subtractions between the absorption spectra of NC-PAH complexes and free NCs, we obtain the absorption spectra of PAHs adsorbed on NC surfaces (Fig. 3c). The absorption spectra of NCA on NC surfaces and in toluene solution are very similar. Interestingly, the spectral features of TCA on NC surfaces are narrower than those of TCA in toluene solution. This suggests TCA molecules likely aggregate in solution, which is expected for extended π-systems and is consistent with previous reports on singlet fission observed for tetracene derivatives in solution[14,38]. The aggregation is also implied by the lack of clear vibronic structures on the PL spectrum of TCA in toluene (Supplementary Fig. 3). In contrast, TCA molecules are well-dispersed in the ligand shell on NC surfaces[39], resulting in narrow line-shapes.

Based on the absorption spectra, the absorption of NCA at 400 nm is essentially zero and the absorption of TCA at 340 nm is more than 20-fold weaker than that of NCs. Thus, the PL spectra of NC-NCA and NC-TCA complexes were acquired by using 400 and 340 nm excitation, respectively, to ensure selective excitation of NCs (Fig. 3d). Compared with free NCs measured

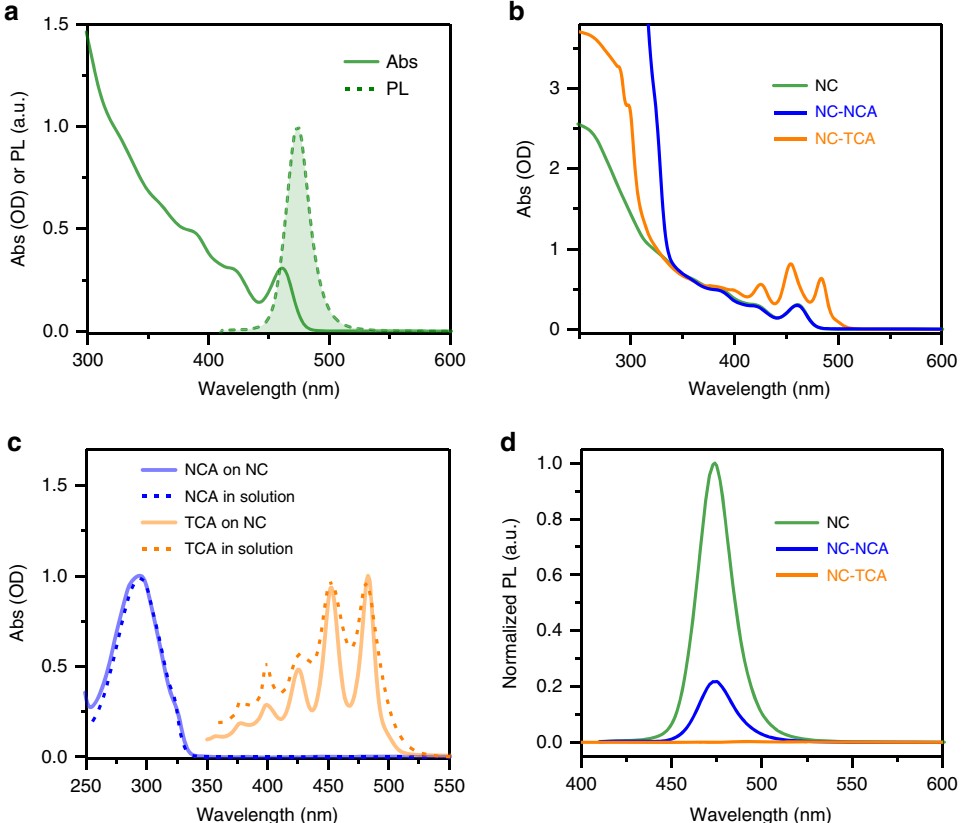

**Fig. 3 Optical properties of the NC-PAH systems. a** Absorption (Abs; solid line) and photoluminescence (PL; dashed line with shading) spectra of CsPbBr$_3$ NCs. **b** Absorption spectra of CsPbBr$_3$ NCs (green), NC-NCA (blue) and NC-TCA (orange) complexes dispersed in hexane. **c** Absorption spectra of NCA and TCA on NC surfaces (blue and orange solid lines, respectively) obtained from spectral differences in (**b**) and absorption spectra of NCA and TCA dissolved in toluene (blue and orange dashed lines, respectively). **d** PL spectra of CsPbBr$_3$ NCs (green) and NC-NCA (blue) and NC-TCA (orange) complexes dispersed in hexane.

under the same experimental conditions, the PL is strongly quenched in both NC-NCA and NC-TCA complexes, with the latter being quenched more efficiently. According to the energetic analysis above, the only quenching pathway in NC*-NCA is TET from NCs to NCA, whereas both TET and hole transfer are possible quenching pathways in NC*-TCA. In addition, because of a strong overlap between the PL spectrum of NCs and the absorption spectrum of TCA, Förster resonant energy transfer (FRET) from NCs to TCA is also allowed. Later we will use time-resolved spectroscopy to clarify the roles of these quenching pathways.

We note that ligand exchange using PAH molecules could, in principle, introduce some new trap states and thus quench the NC emission. In order to examine this possibility, we measured the PL spectra of NC-NCA complexes with large-size NCs (PL peak at ~492 nm). TET from these NCs to NCA is inefficient for lack of enough driving force and/or electronic coupling[32,33]. As shown in Supplementary Fig. 5, PL quenching is negligible for these complexes with large-size NCs. Because the surface chemistry of different-sized NCs prepared using the same synthetic method should be very similar, this observation serves a control to rule out the possibility of PL quenching due to trap states generated in ligand exchange.

**Exciton dynamics in CsPbBr$_3$ NCs.** As mentioned above, one of the major motivations for using highly emissive perovskite NCs as the triplet donor is that carrier trapping, if any, would not significantly complicate the TET dynamics. We first investigated

the exciton/carrier dynamics in pristine CsPbBr$_3$ NCs using a combination of transient absorption (TA) and time-resolved PL (TR-PL) techniques, both having sub-ps time resolution; see Methods. In all these measurements, the average exciton number per NC ($<N>$) was maintained $\ll 1$ such as to exclude multi-excitonic effects and the samples were vigorously stirred to minimize any photocharging effects.

Figure 4a presents representative TA spectra of CsPbBr$_3$ NCs at selected time delays from 3 ps to 40 ns following excitation by a 400 nm laser pulse. The exciton bleach (XB) feature at ~460 nm arises from state-filling effects of both the electron and the hole[40,41], and the photoinduced features at higher energy have been attributed to either biexcitonic interaction induced Stark effect like signals or exciton-activated forbidden transitions[42]. The contributions of the electron and the hole to the XB are determined to be ~75% and 25%, respectively, based on the NC-TCA experiment to be described below. These values are slightly different from those reported for weakly-confined perovskite NCs (~67 and 33%)[40,43,44] likely because the strong quantum confinement effect here modifies electron and hole densities of states in different ways.

The TA kinetics probed at the XB displays only a slight decay within 200 ps and the major decay occurs on the ns timescale (Fig. 4b). In contrast, the TR-PL kinetics probed at the band edge exciton emission peak has a sizable decay within 200 ps, after which the TA and TR-PL kinetics are in general agreement with each other. Since TA probes the contribution-weighted sum of the electron and the hole and PL probes the emission from electron-hole pairs, their difference indicates different trapping behaviours

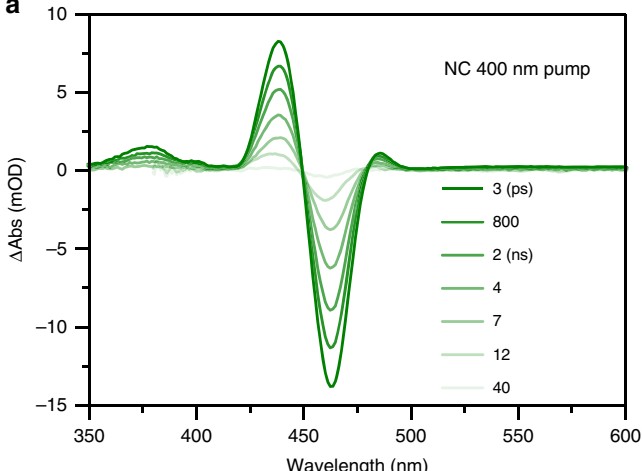

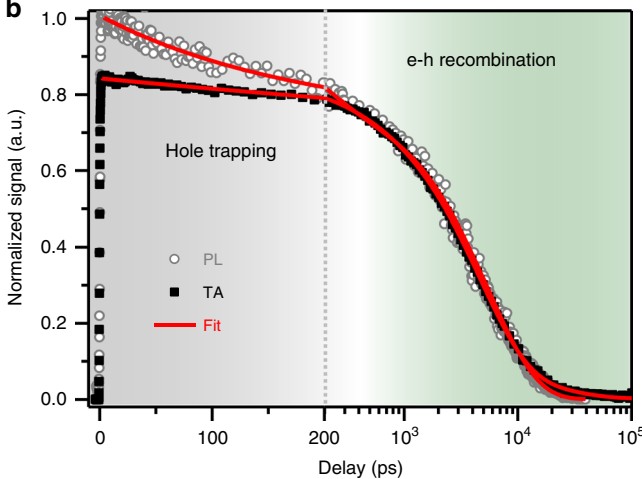

**Fig. 4 Exciton dynamics in free NCs. a** TA spectra of CsPbBr$_3$ NCs at indicated time delays following the excitation by a 400 nm pulse. **b** TA kinetics of NCs probed at the XB centre (~460 nm; black solid squares) and TR-PL kinetics probed at the PL peak (~475 nm; grey open circles). The red solid lines are their fits using a model accounting for a minor hole trapping component and a major electron-hole recombination component. Note that the first 200 ps is plotted in linear scale and the rest in logarithmic scale.

for the electron and the hole. On the basis of the weaker contribution to the XB by the hole, the simplest yet most plausible model is that the PL decay occurring within 200 ps arises from a sub-ensemble of NCs exhibiting ultrafast hole trapping kinetics. By simultaneously fitting the TA and TR-PL kinetics using this model (Supplementary Note 2 and Supplementary Table 2), we identify that ~22% of the NCs has a hole trapping time constant of 160 ± 10 ps, whereas the remaining population (~78%) yields an electron-hole recombination constant of 5.0 ± 0.2 ns. For the former, hole trapping should lead to a long-lived electron component in the TA difference spectra, which is indeed detected in the XB feature (from ~10 to 100 ns; Fig. 4b). The 5.0 ns component is believed to be dominated by electron-hole radiative recombination because its amplitude (78%) is in reasonable agreement with the PL QY of the NC sample (70%) and because its time constant is similar to the lifetime of thiocyanate-treated CsPbBr$_3$ NCs with near-unity QY[45].

**Hole transfer mediated TET**. After determining the exciton dynamics in free NCs, we investigated the mechanism for TET from photoexcited NCs to TCA where hole transfer is

energetically allowed (with a driving force of ~0.3 eV; Fig. 2), also using a combination of TA and TR-PL. Figure 5a shows representative TA difference spectra of NC-TCA complexes at selected delays following excitation by a 340 nm laser pulse which selectively pumped the NCs. The difference spectra are dominated by the XB feature of NCs which exhibits a faster recovery than free NCs. At very long delay times (~30 μs), photoinduced absorption features corresponding to the triplets of tetracene ($^3$TCA*)[28,46] are clearly detected, confirming triplet sensitization or TET from NCs to TCA. However, detailed kinetic analysis is required to uncover the interplay between TET, FRET, hole transfer and hole trapping.

The TA kinetics of NC-TCA complexes probed at the XB shows an ultrafast decay within 50 ps with a relative amplitude of ~25%, while on a similar timescale the TR-PL exhibits complete decay (Fig. 5b). On the basis of the energetics analysis for NC-TCA above, these observations are most consistent with ultrafast hole transfer from NCs to TCA to form the NC$^-$-TCA$^+$ charge separated state and, accordingly, the electron and hole contributions to the XB are 75% and 25%, respectively. The other two energetically allowed pathways, FRET and direct TET, should lead to complete decay of both the XB and the TR-PL rather than partial decay of the XB and complete decay of the TR-PL, as energy transfer should simultaneously annihilate both the electron and the hole. For the FRET pathway, in particular, we should also expect complete decay of the XB followed by gradual formation again of the XB due to energetically allowed back electron transfer from TCA to NCs; such a peculiar kinetic behaviour is clearly not consistent with our experimental observations. The dominance of hole transfer over FRET and direct TET is consistent with previous reports on related systems[43,44]. Specifically, the FRET time constant estimated for a similar system was 0.46 ns[43], which is much slower than the hole transfer process observed here.

During the hole transfer process, we should expect the formation of a ground-state bleach (GSB) feature of TCA. This observation, however, is hindered by a spectral overlap between the GSB of TCA and XB of NCs and by a ~100-fold difference between the extinction coefficients of NCs (~878,000 M$^{-1}$ cm$^{-1}$ at 460 nm) and ground-state TCA (~7400 M$^{-1}$ cm$^{-1}$ at 482 nm). Rather, we observed photoinduced absorption features in the near-IR (~900 and 1150 nm) gradually emerging in ~50 ps (Supplementary Fig. 6). The 900 nm feature is consistent with previously-reported cation absorption peak of triisopropylsilyl-ethynyl tetracene carboxylic acid (TIPS-TCA) at ~930 nm[46]. Moreover, its formation kinetics is consistent with the decay of the XB within 50 ps. Thus, the 900 nm feature can be assigned to the cation radical of TCA (TCA$^+$). The 1150 nm feature has not been reported in previous studies but should also be assigned to TCA$^+$ on the basis of its similar kinetics as the 900 nm one (Supplementary Fig. 6). Both 900 and 1150 nm features were observed also for another two NC-TCA samples with different NC sizes (Supplementary Fig. 7), further confirming their assignment to TCA$^+$. Note that although the cation and anion absorption signals were reported to be very similar[47], we do not consider electron transfer from NCs to TCA here because its driving force is ~0.7 eV less than that of hole transfer (Fig. 2) and because the XB signal shows 25% decay that is more consistent with the hole contribution.

From 50 ps to the ns timescale, the remaining XB component corresponding to the electron decays away and the $^3$TCA* signal, probed at ~485 nm where the NC signal is close to zero, grows in with time (Fig. 5b). Meanwhile, the TCA$^+$ TA signal in the near-IR decays (Supplementary Fig. 6). These observations are fully consistent with the conversion of NC$^-$-TCA$^+$ to NC-$^3$TCA* via electron transfer recombination as suggested to occur in PbS NCs surface anchored with TIPS-pentacene carboxylic acid[26]. The

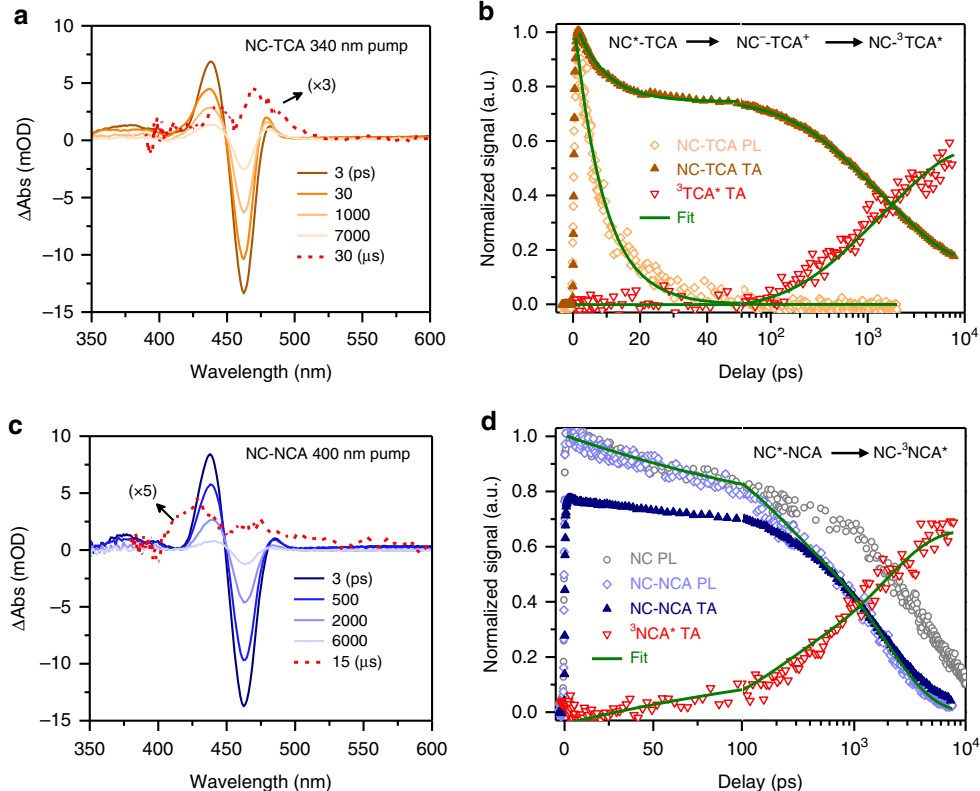

**Fig. 5 TET dynamics in NC-PAH complexes. a** TA spectra of NC-TCA complexes at indicated time delays following the excitation by a 340 nm pulse. The spectrum at 30 μs is amplified by a factor of 3. **b** TA kinetics of NC-TCA complexes probed at the XB centre (~460 nm; dark orange solid triangles) and at the $^3$TCA* (~485 nm; red open triangles) and TR-PL kinetics probed at the PL peak (~475 nm; light orange open diamonds). The green solid lines are their fits using a model described by the inset equation. Note that the first 50 ps is plotted in linear scale and the rest in logarithmic scale. **c**, **d** Similar plots as (**a**, **b**) for NC-NCA complexes. The excitation wavelength is 400 nm and the $^3$NCA* TA feature is probed at 415 nm.

spin of the injected electron is parallel to that of the electron remaining the highest occupied molecular orbital (HOMO) of TCA, thus forming a triplet state, which is also energetically favoured (with a driving force of ~1.3 eV; Supplementary Note 1 and Supplementary Table 1). The sensitized $^3$TCA* states are long-lived (~124 μs) in deoxygenated hexane solutions (Supplementary Fig. 8).

By fitting the TA and PL kinetics in Fig. 5b according to the hole transfer mediated TET model described above (Supplementary Note 2 and Supplementary Table 3), we obtain a hole transfer time constant of 8.9 ± 0.8 ps, similar to previous reports on perovskite NC-tetracene or pentacene systems[43,44], and an averaged electron transfer time constant of 1.9 ± 0.1 ns. That electron transfer is ~200-fold slower than hole transfer can be rationalized by a simple statistical consideration. Specifically, hole transfer is accelerated by the availability of multiple acceptors (~60) as the rate should scale approximately with number of adsorbed acceptors[48], whereas there is only one TCA$^+$ acceptor for the ensuing electron transfer process under the experimental light excitation conditions. Accounting for this factor, the hole and electron transfer time constants per acceptor is ~0.53 and 1.9 ns, respectively. It is also interesting to note that while the lifetime of the NC$^-$-TCA$^+$ charge separated states is only ~1.9 ns, it is as long as ~5.1 μs in our previous report using CsPbCl$_x$Br$_{3-x}$ NCs[43]. This is likely due to a NC size effect. As shown in Supplementary Fig. 9, both hole transfer and electron transfer slow down substantially with increasing CsPbBr$_3$ NC sizes, because both the energetics and electronic coupling terms involved in charge transfer from quantum-confined NCs depend sensitively on NC sizes[48]. For 8.5-nm CsPbBr$_3$ NCs, the electron transfer time (i.e.,

charge separated state lifetime) starts to approach that of the 10-nm CsPbCl$_x$Br$_{3-x}$ NCs studied previously [43].

Because hole transfer outcompetes not only electron-hole recombination (5 ns) but also hole trapping (160 ps), the hole transfer yield is ~98.6%. Furthermore, because of negligible trapping processes competing with the ensuing electron transfer, the electron transfer yield should be near unity, provided that the spin-flip time of the electron in CsPbBr$_3$ NCs is fast enough to ensure that the electron can always "adjust" its spin to form a triplet with the electron in the HOMO of TCA. In this case, the overall TET yield is determined by the hole transfer yield. Indeed, the triplet formation yield, estimated using the maximum signal amplitudes of $^3$TCA* absorption and NC bleach and the extinction coefficients reported for TCA triplets[1] and NCs[49] (Supplementary Note 3), is ~94.8%, similar to the TET yield determined from kinetic parameters. We note that the very weak $^3$TCA* signal observed in Fig. 5a is simply a result of the ~20-fold difference between the extinction coefficients of $^3$TCA* and NCs.

**"Direct" or virtual charge-transfer mediated TET.** In the case of NC-NCA complexes, hole transfer is energetically uphill by ~0.4 eV (Supplementary Note 1 and Table 1). As such, we expect a different TET/triplet sensitization mechanism. Figure 5c shows the TA difference spectra for NC-NCA complexes obtained using 400 nm excitation which also selectively excites the NCs. These spectra illustrate the decay of the NC excited state features and the formation of naphthalene triplets ($^3$NCA*), indicative of TET from NCs to NCA. Comparing the TR-PL kinetics of NCs and NC-NCA complexes in Fig. 5d reveals that the TET occurs

primarily from NCs without trap states (i.e., TET is negligible within 200 ps). The decay of PL occurs on the ns timescale and is consistent with the TA kinetics probed at the XB and also with the growth kinetics of the $^3$NCA* species (Fig. 5d). The kinetic agreement of these three distinct spectroscopic signals suggests direct transfer of electron-hole pairs from the NCs to the triplet state of NCA. Consistently, neither NCA cation nor anion signals were detected in the NIR TA spectra (Supplementary Fig. 10), excluding the channels of TET mediated by real charge separated states.

Fitting the kinetics in Fig. 5d revealed an averaged TET time of $2.1 \pm 0.1$ ns and a TET yield of ~65.0% (Supplementary Note 2 and Supplementary Table 4). The TET time constant per acceptor is ~126 ns. The calculated TET yield is consistent with the NCA triplet formation yield (~64.3%) estimated from TA signal amplitudes (Supplementary Note 3). This yield is much lower than that of NC-TCA (98.6%) mainly because the slow TET process cannot compete with ultrafast hole trapping in a sub-ensemble of NCs. This comparison indicates that in many cases step-wise, CT-mediated TET is a relatively more effective strategy for triplet sensitization as it can compete with other charge trapping or recombination pathways using a fast CT step. Note that, however, this sensitization scheme is often associated with a large energy loss; for example, the energy loss is ~1.4 eV for the NC-TCA system whereas it is only ~0.1 eV for the NC-NCA system.

While the kinetics data for NC-NCA complexes are consistent with a Dexter-type, "direct" TET model originally proposed[10], we do not exclude the possibility that it is also mediated by virtual CT states. This is the so-called "through-configuration" pathway originally proposed in ref. [50]. By analysing the coupling matrix elements, Harcourt et al. found that the Dexter exchange integral for simultaneous two-electron transfer was much weaker than the through-configuration integrals[50]. For our NC-NCA system, the virtual CT state is more likely to be NC$^-$-NCA$^+$ than NC$^+$-NCA$^-$, as the energy detuning between NC$^-$-NCA$^+$ and NC*-NCA is ~2.75-fold weaker than that between NC$^+$-NCA$^-$ and NC*-NCA (0.4 vs. 1.1 eV; Fig. 2). The virtual CT-mediated TET model can borrow further support from the extensive literature on singlet fission (SF). In many ways TET and SF are similar: both are spin-conserving processes and both can proceed via either CT states or direct two-electron transfer. For SF, there is now a general consensus that it is often mediated by CT states[51–55]. In pentacene, for example, the electronic coupling calculated by including CT states is ~4-fold stronger than the direct coupled case despite that their energy is ~0.3–0.6 eV above the singlets[53] and the calculated rates for CT-mediated SF are in good agreement with experimental rates [52].

**Unified picture.** Through a side-by-side comparison between NC-TCA and NC-NCA systems, we propose a unified picture for the models of TET across the inorganic NC/organic molecule interface, as summarized in Fig. 6. For the NC-TCA system, because of a strong electronic coupling ($|V|^2$) between NC*-TCA and NC$^-$-TCA$^+$ states and because of a sufficient driving force ($-\Delta G$), hole transfer is kinetically favoured, as predicted by the non-adiabatic Marcus charge-transfer equation shown below[56]:

$$k = \frac{2\pi}{\hbar} |V|^2 \frac{1}{\sqrt{4\pi\lambda k_B T}} \exp\left[-\frac{(\lambda + \Delta G)^2}{4\lambda k_B T}\right] \quad (1)$$

where $\hbar$ is the reduced Planck constant, $\lambda$ the reorganization energy, $k_B$ the Boltzmann constant and $T$ the temperature. The resulting NC$^-$-TCA$^+$ CT states mediate the ensuing electron transfer step, completing the TET process (Fig. 6a). For the NC-NCA system, both electron and hole transfer are highly endothermic and hence real CT states are not detectable on spectroscopy. However, the electronic coupling ($|V|^2$) between

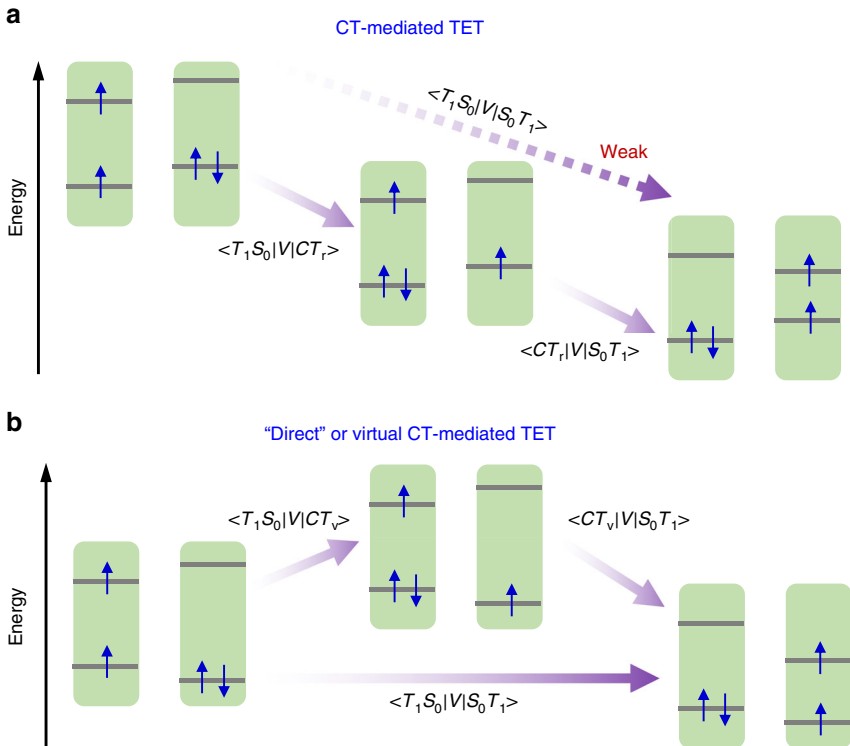

**Fig. 6 TET models. a** If CT is energetically favoured, a real CT state ($CT_r$) mediates TET from the initial ($T_1S_0$) to the final ($S_0T_1$) states. Direct TET from $T_1S_0$ to $S_0T_1$ is avoided due to a relatively weak electronic coupling between them. **b** If CT is energetically disallowed, TET can directly proceed from $T_1S_0$ to $S_0T_1$. However, this "direct" TET could also be mediated a high-energy, virtual CT state ($CT_v$).

NC*-NCA and NC-³NCA* states would be strongly enhanced by the mediation of the NC⁻-NCA⁺ virtual CT states (Fig. 6b).

Our proposed models for TET, in addition to sharing similar physics with SF as mentioned above, are also reminiscent of the hopping versus superexchange models for long-range charge transfer in donor–bridge–acceptor systems[57,58]. In these instances, the hopping model corresponds to the case of real charge separation between the donor and the bridge, whereas in the superexchange model charge transfer from the donor to the acceptor is mediated through high-energy, virtual CT states involving the bridge.

## Discussion

To summarize, in order to clarify the mechanism of triplet energy transfer (TET) across the semiconductor NC/organic interface, we constructed model systems comprising highly emissive lead halide perovskite NCs as the triplet donor and PAHs of either tetracene or naphthalene as the acceptors and studied interfacial dynamics using a combination of ultrafast transient absorption and time-resolved photoluminescence measurements. Our results clearly identify that in the case of NC-tetracene, where hole transfer from NCs to tetracene is energetically favoured, TET is mediated by a real charge separated state, whereas "direct" TET is observed for the NC-naphthalene system because charge-transfer processes are energetically uphill. However, we suspect that "direct" TET can also be mediated through a high-energy, virtual charge-transfer state. The revealed TET mechanisms not only suggest fundamental similarities between TET, singlet fission as well as other charge/energy transfer processes, but also pave the way for rationally designing the energetics at the inorganic semiconductor/organic molecule interface for high-efficiency TET important for many emerging applications.

## Methods

**Chemicals**. The following reagents were used to prepare perovskite nanocrystal-polycyclic aromatic hydrocarbon (NC-PAH) complexes: cesium carbonate ($Cs_2CO_3$, Sigma-Aldrich, 99.9%), lead(II) bromide ($PbBr_2$, Alfa Aesar, 98+%), zinc bromide ($ZnBr_2$, Alfa Aesar, 99.9%), oleic acid (OA, Sigma-Aldrich, 90%), oleylamine (OAm, Acros Organics, 80–90%), 1-octadecene (ODE, Sigma-Aldrich, 90%), 1-naphthalene carboxylic acid (NCA, Alfa Aesar, 98%); 5-tetracene carboxylic acid (TCA) was synthesized as described previously[43].

**Synthesis of CsPbBr₃ NCs**. $CsPbBr_3$ NCs were synthesized by using a hot injection approach with modifications[59]. The synthesis started with preparation of Cs-oleate precursors. 0.25 g $Cs_2CO_3$, 0.8 g oleic acid (OA) and 7 g 1-octadecene (ODE) were loaded into a 50 mL 3-neck flask and vacuum-dried for 1 h at 120 °C using a Schlenk line. The mixture was heated under Ar atmosphere to 150 °C until all the $Cs_2CO_3$ powders were dissolved. The Cs-oleate precursor solution was kept at 100 °C to prevent precipitation of Cs-oleate out of ODE. In another 250 mL 3-neck flask, the precursor solution of Pb and Br was prepared by dissolving 0.45 g $PbBr_2$ and 1.10 g $ZnBr_2$ in a mixture of OA (16 mL), OAm (16 mL) and ODE (26 mL). Then, the precursor solution was vacuum-dried for 1 h at 110 °C, and heated under Ar atmosphere to 140 °C until all the powders were dissolved. After the solution was cooled and kept at 110 °C, 2.4 mL of Cs precursor was injected to initiate the reaction. The reaction was quenched after 6 mins by cooling the flask in an ice bath. The product was centrifuged at 4000 rpm for 15 min to remove the unreacted salts as the precipitate, and the perovskite NCs dispersed in the supernatant were collected. 10 mL of acetone was directly added into the supernatant to precipitate the NCs followed by centrifuging at 4000 rpm for 5 mins. The dried NCs were collected and dissolved in hexane. NCs of different sizes were obtained by tuning the reaction temperature and reaction time, as detailed in refs. [32,60,61].

**Preparation of NC-PAH complexes**. The NC-PAH complexes were prepared by adding PAH (NCA or TCA) powders into a NC solution in hexane, followed by sonication for controlled time (~5 min). The mixture was filtered to obtain a clear solution containing NC-PAH complexes; because the solubility of carboxyl-functionalized PAHs in hexane is negligible, all the PAHs were believed to be anchored to NC surfaces. On the basis of the extinction coefficients of 3.8-nm $CsPbBr_3$ NCs and NCA and TCA, on average ~60 PAH molecules were bound to each NC.

**Cyclic voltammetry**. Cyclic voltammetry (CV) measurements were conducted on a Model CHI700e electrochemical analyser with a three-electrode system under inert gas atmosphere. For TCA, an anhydrous dichloromethane solution containing tetra-*n*-butylammonium hexafluorophosphate (0.1 M) was used as the electrolyte. For NCA, an anhydrous acetonitrile solution containing tetra-*n*-butylammonium hexafluorophosphate (0.1 M) was used as the electrolyte. For the $CsPbBr_3$ NCs, a mixture of solvents (acetonitrile and toluene (1:4 v/v)) containing tetra-butylammonium perchlorate (0.1 M) was used as the electrolyte[36]. Glassy carbon, Pt-wire and Ag/AgCl were used as the working, counter and reference electrodes, respectively. Prior to use, the working electrode was polished over 0.5 μm alumina powder and rinsed with Milli-Q water. The CVs were performed at a scan speed of 100 mVs⁻¹. The CV curves were calibrated with the ferrocene/ferrocenium (Fc/Fc⁺) redox couple as a standard measured under the same conditions. The energy level of Fc/Fc⁺ was taken to be −4.8 eV with respect to vacuum.

**Transient absorption**. The femtosecond pump-probe TA measurements were performed using a regenerative amplified Ti:sapphire laser system (Coherent; 800 nm, 70 fs, 6 mJ/pulse and 1 kHz repetition rate) as the laser source and a Femto-100 spectrometer (Time-Tech LLC). Briefly, the 800 nm output pulse from the regenerative amplifier was split in two parts with a 50% beam splitter. The transmitted part was used to pump a TOPAS Optical Parametric Amplifier (OPA) which generated a wavelength-tunable laser pulse from 250 nm to 2.5 μm as pump beam. The reflected 800 nm beam was split again into two parts. One part with <10% was attenuated with a neutral-density filter and focused into a crystal to generate a white light continuum (WLC) used for probe beam. The probe beam was focused with an Al parabolic reflector onto the sample. After the sample, the probe beam was collimated and then focused into a fibre-coupled spectrometer with CMOS sensors and detected at a frequency of 1 KHz. The intensity of the pump pulse used in the experiment was controlled by a variable neutral-density filter wheel. The delay between the pump and probe pulses was controlled by a motorized delay stage. The pump pulses were chopped by a synchronized chopper at 500 Hz and the absorbance change was calculated with two adjacent probe pulses (pump-blocked and pump-unblocked). The samples were placed in 1 mm airtight cuvettes in a $N_2$-filled glove box and measured under ambient conditions. Nanosecond TA was performed with the EOS spectrometer (Ultrafast Systems LLC). The pump beam is generated in the same way as the femtosecond TA experiment described above. A different white light continuum (380–1700 nm, 0.5 ns pulse width, 20 kHz repetition rate) was used, which was generated by focusing a Nd: YAG laser into a photonic crystal fibre. The delay time between the pump and probe beam was controlled by a digital delay generator (CNT-90, Pendulum Instruments).

**Time-resolved PL**. For an 8 ns time window, the PL decay was measured using a time-resolved fluorescence upconversion set-up (Chimera, Light conversion) and a Pharos laser (1030 nm,100 kHz, 230 fs pulse-duration; Light conversion). Briefly, the fundamental 1030 nm laser pulse was split into two parts. One part was used to pump a TOPAS OPA to generate wavelength-tunable excitation pulses; the other was used as the gate pulse. The emitted light was collected by lens and focused into a BBO crystal together with the 1030 nm gate pulse to generate the up-converted signal via sum-frequency-generation (SFG). The up-converted photons were focused onto the entrance slit of a monochromator and then detected by the spectrometer. The fluorescence decay curve was obtained by delaying the gate pulse using a mechanical delay stage. For a 100 ns time window, the time-resolved PL decay was measured using time-correlated single-photon counting (TCSPC) set-up, which uses the same excitation source and camera as the fluorescence upconversion set-up and has a temporal resolution of 100 ps. All samples were placed in 1 mm airtight cuvettes in a $N_2$-filled glove box and were vigorously stirred in all the measurements.

## Data availability
The experiment data that support the findings of this study are available from the corresponding author upon reasonable request.

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

## Acknowledgements

K.W. acknowledges financial support from the Ministry of Science and Technology of China (2018YFA0208703), the National Natural Science Foundation of China (21975253), the Strategic Pilot Science and Technology Project of Chinese Academy of Sciences (XDA21010206) and the LiaoNing Revitalization Talents Program (XLYC1807154). X.L. acknowledges financial support from the National Natural Science Foundation of China (21903084) and The Dalian Institute of Chemical Physics (DICP I201914). F.N.C. acknowledges support from the Air Force Office of Scientific Research (FA9550-18-1-0331). This paper is dedicated to Dalian Institute of Chemical Physics, CAS, on the occasion of its 70th anniversary.

## Author contributions

K.W. conceived the ideas and designed the project. X.L., Y.H., Y.L., X.L. and T.D. synthesized the samples. X.L., Z.C. and G.L. performed spectroscopy experiments. X.L. and K.W. analysed the data. X.L., C.N. and M.W. performed CV measurements. K.W. and F.N.C. formulated the possible exciton transfer processes occurring at the interfaces. K.W. wrote the paper with contributions from all authors.

## Competing interests

The authors declare no competing interests.
