## [Peer Review File · Nature Communications]

Reviewers' comments:

Reviewer #1 (Remarks to the Author):

In this manuscript, Luo et al. present experiments aimed at understanding triplet energy transfer (TET) from photoexcited CsPbBr₃ nanocrystals (NCs) to tetracene or naphthalene (TCA or NCA) ligands attached to the NC surface. The authors find that the two complexes rely upon different mechanisms for the TET, due to differences in interfacial energetics – in NC-TCA complexes, TET proceeds via a real hole-transfer mediated TET whereas in NC-NCA complexes, TET proceeds through a virtual hole-transfer state. The latter mechanism is based upon analysis of the coupling matrix element, since the data are also consistent with a direct TET mechanism. The manuscript is well written and is laid out nicely with regards to the discussion of data and corresponding development of the mechanistic picture. The authors are clear when they speculate about certain possibilities and do a nice job connecting each experimental result to the mechanism. The topic is also certainly of current interest to the community of scientists exploring singlet fission and triplet energy/charge transfer in hybrid systems, so I think it is a good fit in Nature Communications. As such, I think the article could be considered for publication in Nature Communications if a few points, discussed in detail below, are addressed.

- The excitation conditions were chosen to selectively excite the NCs. The authors should include the absorbance spectra of the neat TCA and NCA molecules, at least in the supporting information, so the reader can confirm that there is no excitation of these molecules under the chosen conditions. If there is some non-negligible excitation at these wavelengths, the authors should discuss what effect(s) this might have on their interpretation.
- The authors suggest that the low number of trap states is an inherent advantage for these NCs, since it simplifies the number of kinetic pathways within the native NCs. I agree with this, but the ligand exchange process could potentially introduce some surface dangling bonds that are not present on the native NCs. Since the PL is highly quenched upon exchange with the TCA and NCA, it would be hard to discern whether this PL quenching is solely from TET or has some component of carrier trapping induced by new defect states. Do the authors have any control experiment with 'inert' ligands showing that the ligand exchange process does not introduce any additional dangling bonds (and associated decays in the TA dynamics) on the NC surface?
- Page 8, line 164 – 168: The authors observe a peak in the NIR that they assign to the TCA cation, helping them to assign the mechanism in this system to a hole transfer-mediated TET. The other piece of information is that the relatively fast kinetics observed in the NC-TCA XB recovery have a ~25% amplitude, close to what the authors estimate for the hole contribution to the XB. The amplitude argument seems sensible qualitatively, but could also be coincidence (see also uncertainty about new trap states in ligand-exchanged NCs discussed above), so I think the second observation (cation spectral signature in the NIR) must be relied upon more heavily. I have a couple of questions about this assignment to the cation NIR absorption.
 - o The assignment to the cation is based on recent literature results where electron transfer from TCA to an acceptor generates the cation. The peak observed in the current study is around 1150 nm, whereas the peak observed in reference 33 (Fig. 4) is around 930 nm (shifted to 1000 nm in some cases due to solvent dielectric screening effects). How do the authors account for such a large discrepancy (150 – 220 nm) between their observations and this literature result? Have the authors performed any spectro-electrochemistry or molecular doping experiments to generate the cation in their own molecules to confirm the assignment?
 - o Also, do such experiments exist in the literature for hole transfer from TCA? In other words, what is the spectral signature of the TCA anion and is it distinct from the cation spectral signature? While the band diagram and the driving forces calculated by the authors make it seem unlikely that the electron transfer would proceed first, if these calculations are off, any experimental evidence more concretely connecting this NIR absorption to the cation (and not anion) would be useful. It seems like a shift of 150 – 220 nm (relative to ref. 33) could be consistent with either

differences in dielectric environment, or could also correspond to anion versus cation, so this is important to consider.

o Finally, I don't see a corresponding figure in the SI for the NC-NCA complex. If the proposed mechanism is correct, since this complex does not proceed through a real charge transfer state, there should be no NIR absorption for NCA cations (or anions). Is this the case? This would serve as an important control experiment for supporting the mechanism.

- Page 10, first paragraph: The authors point to two possibilities for why the electron transfer proceeds much more slowly than hole transfer in the TCA system. The authors are clear that the possibilities are speculative, so I have no problem with this discussion, but I have a question for the second suggestion. The second possibility mentioned is an Auger-mediated hole transfer where the hole imparts energy to the electron by exciting it higher into the conduction band. This assumes that the availability of conduction band states significantly exceeds the availability of valence band states. Is there any theoretical evidence for this from a calculated band structure perspective for these (or similar) perovskite NCs?

Reviewer #2 (Remarks to the Author):

Energy transfer systems employing inorganic nanocrystals interfaced with organic spin-triplet energy acceptors have come into vogue for a range of applications, including photon upconversion materials and singlet fission-based solar cells. A variety of mechanisms for triplet transfer across nanocrystal-molecule junctions have been proposed in different systems, highlighting that there is likely a distribution of pathways along which energy transfer from one material to the other depending on the chemical structure of the junction. The authors' goal is to identify some simple principles to a priori predict which particular energy transfer mechanism should operate in a given system.

Experimentally, the authors study perovskite nanocrystals functionalized with either naphthalene or tetracene triplet energy acceptors and show that photoexciting the nanocrystal in each system drives triplet energy transfer to the organic acceptor, but via different mechanisms. While energy transfer to naphthalene involves a single Dexter energy transfer step, triplet transfer to tetracene appears to occur via a charge separated intermediate. To explain why these two related systems exhibit different energy transfer mechanisms, the authors formulate a simplistic model based on the redox properties of the nanocrystal-molecule junction.

Overall, the picture presented by the authors seems self-consistent. However, this work overlaps substantially with recent work published by the authors this year in *J. Phys. Chem. Lett.* (Ref. 28, main text) and *Chem. Sci.* (Ref. 32, main text). The naphthalene data appears to be nearly identical to that published in *J. Phys. Chem. Lett.* while the tetracene data is similar to that published in *Chem. Sci.* but is a bit puzzling in that it shows different results as only charge separation was seen in the *Chem. Sci.* work with no subsequent triplet formation. The discrepancy between these two studies could reflect the difference in the bandgap of the perovskite nanocrystals employed in each. However, I don't think this should be the case as a wider bandgap perovskite nanocrystal was used in the *Chem. Sci.* work, which should provide a larger driving force for triplet energy transfer.

While the experimental work carries less impact due to its appearance elsewhere, the theoretical approach to compute driving forces for charge separation and triplet energy transfer is new and offers a potential means to rectify the different dynamics the authors observe in their two systems. However, this model needs some additional vetting before publication as its accuracy in computing excitonic energy levels are off relative to experiment to a large enough level that I am not sure its mechanistic predictions can be trusted for the systems the authors investigate. In particular, I suspect that it would predict that the tetracene system investigated in the authors' prior *Chem.*

Sci. work would also go on to form triplet excitons. While I appreciate the authors' approach and think it is well thought out, one would like to see it vetted using experimental values for redox properties measured by electrochemical, spectroscopic, and photoelectron methods rather than using purely computed values.

Overall, I feel the experimental data alone does not merit publication in Nature Communications. However, if packaged with a more convincing theoretical model, vetted by experimental data rather than computed energy levels, which is shown to not only explain data in this publication but also work the authors and other groups have published in looking at perovskite to acene triplet energy transfer, then I think this manuscript could carry the import needed for publication in Nature Communications.

Below, I identify additional specific issues for the authors to address while revising their manuscript.

1) A key issue I have with this manuscript is the vetting of the energy level diagram shown in figure 1c. This picture serves as the basis for the authors' conclusion that tetracene is capable of accepting a hole from a perovskite nanocrystal but naphthalene is not. However, I question the accuracy of the energy levels pictured. For example, the energy levels shown for the perovskite imply a bandgap of ~ 3 eV, but the optical spectra shown in figure 1d right below show a band edge exciton transition at ~ 2.7 eV. These levels also appear to be shifted down with respect to vacuum by ~ 0.3 eV relative to values reported by the authors in their Chem. Sci. paper. These bandshifts are troubling given that the authors point to barrier of only ~ 30 meV (on the order of $k_B T$) to argue as to why hole transfer between the perovskite valence band and naphthalene does not occur. Energy levels computed for tetracene appear to be equally off, which is computed to have a band gap of 2.3 eV, but appears to have one closer to 2.6 eV looking at optical spectra in figure 1d.

Do the authors have any means available to experimentally vet the energy levels shown in figure 1c such as CV or photoelectron spectroscopy? Without such data, I find difficult to believe that the redox properties of the two materials alone is enough to drive the difference in dynamics they observe between their two systems. Providing such experimental vetting is key as the energy level diagram in figure 1c forms the crux for their conclusions.

2) It is clear the authors have two different kinetic models in mind for describing both the dynamics of their tetracene:perovskite system (perovskite exciton \leftrightarrow charge separated state \leftrightarrow tetracene triplet) and tetracene:naphthalene system (perovskite exciton \leftrightarrow naphthalene triplet). I am a bit puzzled then as to why the authors fit their kinetic data to a series of single exponential or biexponential functions at specific probe wavelengths rather than using a global analysis package to fit their full transient spectra data sets to the solution of a coupled set of kinetic equations. This fit could be applied simultaneously to both their visible and near-infrared spectral data and would provide a more natural interpretation of the rate constants recovered from their model.

3) Have the authors characterized the average number of tetracene/naphthalene molecules that bind to each perovskite nanocrystal? If these values are known, then presumably the rate constants the authors extract from their data could be scaled to account for the fact that multiple molecules bind to each nanocrystal, thereby accelerating rates measured for energy transfer. Presumably, the reported rates for triplet/charge transfer are enhanced to a large degree by the number of acceptors.

4) Is there any evidence for modification of the absorption features of tetracene/naphthalene upon binding to the perovskite? Looking at the absorption onset for tetracene in particular (figure 1d), there appears to be a weak shoulder that extends out past 500 nm that could indicate some sign of aggregation of tetracene monomers on the surface.

5) The authors estimate that holes contribute to 25% of the XB photobleach of the perovskite nanocrystals, but I find the data in supplementary figure 2 from which this conclusion is drawn very confusing. What I would expect to see in the XB dynamics shown in panel b would be multiexponential decay kinetics, with an initial rate corresponding first to electron transfer to rhodamine B followed by a separate slower decay due to charge recombination. The electron transfer step should be reflected by a bleaching of rhodamine B followed by a decay of the rhodamine bleach as charge recombination occurs. While it does appear that the growth of the rhodamine bleach tracks the initial portion of the perovskite bleach decay, after ~ 1 ns the rhodamine bleach plateaus while the perovskite bleach continues to decay. There also doesn't appear to be multiple decay timescales that can be readily seen in the perovskite bleach recovery. This makes me question if the amplitude of the decay can fully be ascribed to only forward and back charge transfer to rhodamine. If another process impacts the perovskite bleach recovery, this would invalidate the author's assignment of 25% of the XB photobleach to the hole.

As a minor followup question, why does the XB photobleach grow for the first ps following photoexcitation? Is this tied to carrier cooling within the perovskite nanocrystal?

6) For the tetracene:perovskite system, I would expect hole transfer to tetracene to lead to a photobleach of tetracene absorption features in the visible spectral range, but evidence of tetracene photobleaching is not readily apparent looking at data shown in figure 3c. Presumably this reflects the difference in extinction between the perovskite and tetracene triplet acceptors which could make the tetracene bleach difficult to observe. However, this extinction ratio is known and the authors should be able to compute the estimated strength of the tetracene photobleach if full each perovskite photoexcitation produced a tetracene cation. I suspect this should give some observable features in the author's transient spectra that could be associated with photoexcited tetracene. If the authors find the tetracene photobleach is estimated to be larger than that observed experimentally, does this perhaps provide some evidence that the hole may not be fully displaced onto the tetracene core?

7) Looking at the optical absorption and emission spectra reported for the perovskite:tetracene system (figures 1d & 1e), I'm a bit surprised that FRET energy transfer from the perovskite to tetracene is not considered as the perovskite emission perfectly overlaps the tetracene absorption. I realize this pathway was previously considered by the authors in their Chem. Sci. paper earlier this year and deemed too slow to be important, but that manuscript reported data on a perovskite with a narrower bandgap that should form a worse FRET pair. Is it possible that the charge transfer step the authors observe first involves formation of a singlet exciton on tetracene followed by charge transfer back to the perovskite? How would the authors rule out such a pathway?

8) Estimates for the triplet extinction of both tetracene and naphthalene are available as well as the tetracene cation, and I would encourage the authors to estimate the yields of producing each of these species from their transient data. Do all excitations eventually go on to form triplets? If not, where are they being lost? Does charge recombination to the ground state in the tetracene system play a major competing role? A naïve conclusion based on the weak triplet induced absorption relative to the perovskite features shown in figure 3 would lead one to conclude that triplet energy transfer is somewhat inefficient in both systems, which would hamper the utility of these materials for applications.

9) On p. 11, the authors make reference to "through-configuration" pathways contributing to the matrix element for triplet energy transfer from a perovskite nanocrystal to naphthalene. This type of contribution has been referred to in the literature by a few different names, including as a "superexchange" contribution, "charge resonance" contribution, and "virtual charge transfer" contribution. It would be worth making use of some of these other terms for clarity as I believe "through-configuration" is not often used.

10) The value reported for the averaged electron transfer time constant (1878 +/- 74 ps) seems very precise given the error and accuracy of other values reported in the manuscript.

11) Minor point, on p. 5 I would state "ground state NCA and TCA" rather than "NCA and TCA ground states".

Reviewer #3 (Remarks to the Author):

Re: Charge transfer mediated triplet energy transfer across the inorganic/organic interface, by Luo, et al.

This work proposes a unified model for triplet energy transfer from nanocrystals to molecular acceptors that is mediated by charge transfer states. The experimental system employs CsPbBr₃ perovskite particles with very high quantum yields (and apparently very little spectral shift between their absorption and emission) which is consistent with the absence of surface states. Combinations of the nanocrystals and naphthalene carboxylic acid and tetracene carboxylic acid are studied using transient absorption and PL. The tetracene and naphthalene derivatives are chosen to energetic allow and disallow, respectively, the formation of intermediate charge transfer states.

Overall, the manuscript seems straightforward and a valuable contribution to the field. The tetracene derivative quenches the nanocrystal PL very rapidly, and the transient and energetic structure appears consistent with a intermediate charge transfer state.

I have some concerns about the NCA experiments. On first examination the energetic structure for the naphthalene derivative interface (Fig. 1c) looks like it could support a charge transfer state via either hole transfer. On closer reading (especially the supplementary), various corrections are proposed that destabilize the hole transfer state. These corrections are not represented in the current figure. I am also concerned about that diagram because it suggests that the oxidation energy of the NCA is the same as the oxidation energy of NCA*, the triplet excited state. It might be helpful to draw a separate diagram for each system with energies of the initial state, CT mediating state, and acceptor triplet.

I found the emphasis on a 'unified picture' to be unhelpful in the NCA section. It seems to me that in one case, energy transfer proceeds via charge separation, and in the other case, it proceeds by Dexter transfer. The authors propose that the virtual CT is very important in the NCA system, but it is unclear what evidence exists to support this proposal. Indeed, if virtual CT states are so important here, wouldn't they also be crucial intermediates in almost every example of triplet energy transfer that is presently called Dexter transfer?

Arguments about the definition of Dexter transfer aside, I found the arguments about charge separation improving the yield to be far more interesting and possibly worthy of highlighting in the abstract etc....

Finally, it might be helpful to plot the absorption and emission of the CsPbBr₃ particles on top of one another. It would be good to confirm that the spectral shift is very small.

Reviewers' comments:

Reviewer #1 (Remarks to the Author):

In this manuscript, Luo et al. present experiments aimed at understanding triplet energy transfer (TET) from photoexcited CsPbBr₃ nanocrystals (NCs) to tetracene or naphthalene (TCA or NCA) ligands attached to the NC surface. The authors find that the two complexes rely upon different mechanisms for the TET, due to differences in interfacial energetics – in NC-TCA complexes, TET proceeds via a real hole-transfer mediated TET whereas in NC-NCA complexes, TET proceeds through a virtual hole-transfer state. The latter mechanism is based upon analysis of the coupling matrix element, since the data are also consistent with a direct TET mechanism. The manuscript is well written and is laid out nicely with regards to the discussion of data and corresponding development of the mechanistic picture. The authors are clear when they speculate about certain possibilities and do a nice job connecting each experimental result to the mechanism. The topic is also certainly of current interest to the community of scientists exploring singlet fission and triplet energy/charge transfer in hybrid systems, so I think it is a good fit in Nature Communications. As such, I think the article could be considered for publication in Nature Communications if a few points, discussed in detail below, are addressed.

Response: We thank the reviewer very much for his/her kind comments on our work. Below we address his/her concerns point-by-point.

- The excitation conditions were chosen to selectively excite the NCs. The authors should include the absorbance spectra of the neat TCA and NCA molecules, at least in the supporting information, so the reader can confirm that there is no excitation of these molecules under the chosen conditions. If there is some non-negligible excitation at these wavelengths, the authors should discuss what effect(s) this might have on their interpretation.

Response: We thank the reviewer for the good suggestion. We have now included the absorption spectra of neat TCA and NCA molecules in a new Fig. 3c, according to which the absorptions of NCA at 400 nm and TCA at 340 nm are indeed negligible compared to CsPbBr₃ NCs at corresponding wavelengths.

Revision: We add the following sentence before we discuss the PL spectra of NC-NCA and NC-TCA complexes:

“Based on the absorption spectra, the absorption of NCA at 400 nm is essentially zero and the absorption of TCA at 340 nm is more than 20-fold weaker than that of NCs.”

A new Fig. 3c is also added to show the absorption spectra of NCA and TCA on NC surfaces and those of neat NCA and TCA dissolved in toluene.

- The authors suggest that the low number of trap states is an inherent advantage for these NCs, since it simplifies the number of kinetic pathways within the native NCs. I agree with this, but the ligand exchange process could potentially introduce some

surface dangling bonds that are not present on the native NCs. Since the PL is highly quenched upon exchange with the TCA and NCA, it would be hard to discern whether this PL quenching is solely from TET or has some component of carrier trapping induced by new defect states. Do the authors have any control experiment with ‘inert’ ligands showing that the ligand exchange process does not introduce any additional dangling bonds (and associated decays in the TA dynamics) on the NC surface?

Response: We thank the reviewer for this very insightful comment. Indeed, there is a possibility that ligand exchange could introduce additional defect states. Thus, a control experiment using “inert” ligand, as suggested by the reviewer, is needed. We actually have this data but they are not presented in our previous submission. We have measured the PL spectra of larger-size NCs (PL peak at ~492 nm) before and adding NCA ligand. Based on our previous work (*J. Phys. Chem. Lett.* **2019**, *10*, 1457-1463), TET from these NCs to NCA is inefficient for lack of enough driving force and/or electronic coupling. In this case, PL quenching is negligible (Fig. R1). Because the surface chemistry of different-sized NCs prepared using the same synthetic method should be very similar, this observation serves a control to rule out the possibility of PL quenching due to trap states generated in ligand exchange.

Fig. R1. (a) Absorption and (b) PL spectra of larger-size CsPbBr₃ NCs (gray) and NC-NCA complexes (red). PL was acquired using 400 nm excitation.

Revision: We add the following paragraph to describe this control experiment: “We note that ligand exchange using PAH molecules could, in principle, introduce some new trap states and thus quench the NC emission. In order to examine this possibility, we measured the PL spectra of NC-NCA complexes with large-size NCs (PL peak at ~ 492 nm). TET from these NCs to NCA is inefficient for lack of enough driving force and/or electronic coupling.^{32,33} As shown in Supplementary Fig. 5, PL quenching is negligible for these complexes with large-size NCs. Because the surface chemistry of different-sized NCs prepared using the same synthetic method should be very similar, this observation serves a control to rule out the possibility of PL quenching due to trap states generated in ligand exchange.”

In the SI, we add Fig. R1 above as a new Supplementary Fig. 5.

• Page 8, line 164 – 168: The authors observe a peak in the NIR that they assign to the

TCA cation, helping them to assign the mechanism in this system to a hole transfer-mediated TET. The other piece of information is that the relatively fast kinetics observed in the NC-TCA XB recovery have a ~25% amplitude, close to what the authors estimate for the hole contribution to the XB. The amplitude argument seems sensible qualitatively, but could also be coincidence (see also uncertainty about new trap states in ligand-exchanged NCs discussed above), so I think the second observation (cation spectral signature in the NIR) must be relied upon more heavily. I have a couple of questions about this assignment to the cation NIR absorption.

The assignment to the cation is based on recent literature results where electron transfer from TCA to an acceptor generates the cation. The peak observed in the current study is around 1150 nm, whereas the peak observed in reference 33 (Fig. 4) is around 930 nm (shifted to 1000 nm in some cases due to solvent dielectric screening effects). How do the authors account for such a large discrepancy (150–220 nm) between their observations and this literature result? Have the authors performed any spectro-electrochemistry or molecular doping experiments to generate the cation in their own molecules to confirm the assignment?

Response: We thank the reviewer for this very insightful comment. Indeed, we have observed both ~900 and 1150 nm features in our NIR-TA data. We reported the signal and kinetics for 1150 nm because the white-light-continuum probe pulse used in our experiment was generated by 800 nm fundamental beam, which led to relatively poor S/N ratio near 800 nm.

We now present wider-range TA spectra showing both 900 and 1150 nm features (Fig. R2). The 900 nm feature is consistent with previously-reported cation absorption peak of triisopropylsilylethynyl tetracene carboxylic acid (TIPS-TCA) at ~930 nm. Moreover, its formation kinetics is also consistent with the decay of the XB within 50 ps (Fig. R2). Thus, the 900 nm feature can be assigned to the cation radical of TCA (TCA⁺).

Fig. R2. (a,b) NIR TA spectra of CsPbBr₃ NCs (a) and NC-TCA complexes (b) at indicated time delays following the excitation by a 340 nm pulse. (c) 2D pseudo-color plot of the NIR TA spectra of NC-TCA complexes. (d) Comparison of the extracted radical kinetics at ~900 (black triangles) and ~1150 nm (blue circles) of TCA and the scaled XB kinetics of NCs (red solid line).

The 1150 nm feature has not been reported in previous studies, likely because it is beyond the detection limit for many spectroelectrochemistry set-ups used in the literature (<1100 nm), but should also be assigned to TCA⁺ on the basis of its similar kinetics as the 900 nm one; as shown in Fig. R2. Both 900 and 1150 nm features were observed also for another two NC-TCA samples with different NC sizes (lowest energy absorption peaks at 440 and 450 nm, respectively; Fig. R3), further confirming their assignment to TCA⁺.

Fig. R3. 2D pseudo-color plots of NIR TA spectra of CsPbBr₃ NCs (a) and NC-TCA complexes (b) with the lowest energy absorption peak at 440 nm. (c,d) The same plots as (a) and (b) but for NCs with the lowest energy absorption peak at 450 nm.

Revision: We revised the content describing the NIR-TA features:

“Rather, we observed photoinduced absorption features in the near-IR (~900 and 1150 nm) gradually emerging in ~50 ps (Supplementary Fig. 6). The 900 nm feature is consistent with previously-reported cation absorption peak of triisopropylsilylethynyl tetracene carboxylic acid (TIPS-TCA) at ~930 nm.⁴⁶ Moreover, its formation kinetics is consistent with the decay of the XB within 50 ps. Thus, the 900 nm feature can be assigned to the cation radical of TCA (TCA⁺). The 1150 nm feature has not been reported in previous studies but should also be assigned to TCA⁺ on the basis of its similar kinetics as the 900 nm one (Supplementary Fig. 6). Both 900 and 1150 nm

features were observed also for another two NC-TCA samples with different NC sizes (Supplementary Fig. 7), further confirming their assignment to TCA⁺.”

In the SI, we add Figs R2 and R3 as new Supplementary Figs 6 and 7, respectively.

Also, do such experiments exist in the literature for hole transfer from TCA? In other words, what is the spectral signature of the TCA anion and is it distinct from the cation spectral signature? While the band diagram and the driving forces calculated by the authors make it seem unlikely that the electron transfer would proceed first, if these calculations are off, any experimental evidence more concretely connecting this NIR absorption to the cation (and not anion) would be useful. It seems like a shift of 150 – 220 nm (relative to ref. 33) could be consistent with either differences in dielectric environment, or could also correspond to anion versus cation, so this is important to consider.

Response: We thank the reviewer for this insightful comment. Because TCA was not photoexcited in our experiment, hole transfer from photoexcited TCA to NCs is not considered here. Rather, a possible pathway for TCA anion generation should be electron transfer from photoexcited NCs to TCA. This process, however, has a driving force that is 0.7 eV less than the driving force for hole transfer from photoexcited NCs to TCA.

Per the reviewer’s suggestion, we looked up for TCA anion signal in the literature and found it was very similar to the cation signal; see Fig. R4 (*J. Am. Chem. Soc.* **1973**, *95*, 3473-3483). However, based on the 0.7 eV difference in electron and hole transfer driving forces and on the 25% decay of the XB feature, we believe that our results are consistent with hole transfer.

Fig. R4. Absorption spectra of tetracene anion (top) and cation (bottom), adapted from *J. Am. Chem. Soc.* **1973**, *95*, 3473-3483. Note that kK is kiloKaiser; 10kK=10000 cm⁻¹=1000 nm.

Revision: We add the following content to rationalize the hole transfer picture:

“Note that although the cation and anion absorption signals were reported to be very similar,⁴⁷ we do not consider electron transfer from NCs to TCA here because its driving force is ~ 0.7 eV (Fig. 2) less than that of hole transfer and because the XB signal shows 25% decay that is more consistent with the hole contribution.”

Finally, I don't see a corresponding figure in the SI for the NC-NCA complex. If the proposed mechanism is correct, since this complex does not proceed through a real charge transfer state, there should be no NIR absorption for NCA cations (or anions). Is this the case? This would serve as an important control experiment for supporting the mechanism.

Response: We thank the reviewer for this very useful suggestion. We have measured the NIR-TA spectra for NC-NCA. As shown in Fig. R5, The spectral shape of NC-NCA complexes is the same as that of free NCs, except that the broad-band photoinduced absorption (PA) band decays faster in the presence of NCA due to TET depleting NC excited states. According to *J. Am. Chem. Soc.* **1973**, *95*, 3473-3483, NCA cation and anion should have absorption features within ~ 800 -900 nm, the absence of these features on NC-NCA spectra confirms that there are not real charge separated states formed in NC-NCA.

Fig. R5. Near-infrared TA spectra of CsPbBr₃ NCs (a) and NC-TCA complexes (b) at indicated time delays following the excitation by a 400 nm pulse.

Revision: We add the following sentence to the discussion of NC-NCA data:

“Consistently, neither NCA cation nor anion signals were detected in the NIR TA spectra (Supplementary Fig. 10), excluding the channels of TET mediated by real charge separated states.”

In the SI, we add Fig. R5 as a new Supplementary Fig. 10.

• Page 10, first paragraph: The authors point to two possibilities for why the electron transfer proceeds much more slowly than hole transfer in the TCA system. The authors are clear that the possibilities are speculative, so I have no problem with this discussion, but I have a question for the second suggestion. The second possibility mentioned is an Auger-mediated hole transfer where the hole imparts energy to the electron by exciting it higher into the conduction band. This assumes that the availability of conduction band states significantly exceeds the availability of valence band states. Is there any theoretical evidence for this from a calculated band structure perspective for these (or similar) perovskite NCs?

Response: We thank the reviewer for this question. There is a key difference between the electron and hole transfer events. During hole transfer, there is an electron at the conduction band edge of the NC that can accept the excessive hole transfer driving and be excited inside the conduction band. In contrast, during the following electron transfer process, there is no hole anymore in the valence band of NCs (it is in TCA already), and hence the Auger-assisted picture does not apply to this electron transfer process. We hope this clarifies the reviewer’s question.

However, because this Auger-assisted picture is not broadly accepted yet in the charge transfer community, we think it might be confusing to the readers, as it does to the reviewer. We decide to remove this statement because it is simply a conjecture. But the first reason, the number of acceptors, should be obvious. We therefore revise our statement to only include the first possibility.

Revision: We simplify this discussion as follows:

“That electron transfer is ~200-fold slower than hole transfer can be rationalized by a simple statistic consideration. Specifically, hole transfer is accelerated by the availability of multiple acceptors as the rate should scale approximately with number of adsorbed acceptors⁴⁸, whereas there is only one TCA⁺ acceptor for the ensuing electron transfer process under the experimental light excitation conditions.”

Reviewer #2 (Remarks to the Author):

Energy transfer systems employing inorganic nanocrystals interfaced with organic spin-triplet energy acceptors have come into vogue for a range of applications, including photon upconversion materials and singlet fission-based solar cells. A variety of mechanisms for triplet transfer across nanocrystal-molecule junctions have been proposed in different systems, highlighting that there is likely a distribution of pathways along which energy transfer from one material to the other depending on the

chemical structure of the junction. The authors' goal is to identify some simple principles to a priori predict which particular energy transfer mechanism should operate in a given system.

Experimentally, the authors study perovskite nanocrystals functionalized with either naphthalene or tetracene triplet energy acceptors and show that photoexciting the nanocrystal in each system drives triplet energy transfer to the organic acceptor, but via different mechanisms. While energy transfer to naphthalene involves a single Dexter energy transfer step, triplet transfer to tetracene appears to occur via a charge separated intermediate. To explain why these two related systems exhibit different energy transfer mechanisms, the authors formulate a simplistic model based on the redox properties of the nanocrystal-molecule junction.

Overall, the picture presented by the authors seems self-consistent. However, this work overlaps substantially with recent work published by the authors this year in J. Phys. Chem. Lett. (Ref. 28, main text) and Chem. Sci. (Ref. 32, main text). The naphthalene data appears to be nearly identical to that published in J. Phys. Chem. Lett. while the tetracene data is similar to that published in Chem. Sci.

Response: We thank the reviewer very much for these comments and for his/her attention to our recent work on related systems. Indeed, we have presented related studies on NC-TCA and NC-NCA in our Chem. Sci. and JPCL paper, respectively. But the key data and analysis and the focus here were very different. Below we elaborate on these differences to justify the novelty of the current work. We hope that the reviewer might find these clarifications useful.

- 1) In the JPCL paper, we simply showed that NCA triplets could be sensitized by CsPbBr₃ NCs of various sizes, but we didn't comment on the detailed mechanism as we have done here. Here we show the decay of the XB TA feature and the decay of time-resolved PL (not measured there) are mostly the same and they are correlated with the formation kinetics of the NCA triplet TA feature (not measured there). Also, per reviewer 1's suggestion, we add the NIR-TA data to show the absence of NCA cation or anion signals (not measured there). These observations, taken together, clearly establish that TET occurs in a "direct" pathway, which makes a clear-cut comparison with the NC-TCA data.
- 2) In the Chem. Sci. paper, we showed that hole transfer from ~10 nm CsPbCl_xBr_{3-x} NCs to TCA occurred on a picosecond timescale and a long-lived charge separated state was obtained. Because the focus there was to use the ultrafast hole transfer to dissociate short-lived multiexcitons in the NCs, we didn't pay attention to triplet sensitization there. In fact, we haven't fully formulated the physical picture of hole transfer mediated TET at all at that time. Now we understand that the charge recombination process we reported in the Chem. Sci. paper is essentially electron transfer from NCs to TCA⁺ to form TCA triplets, which we will elaborate below. In the current work, we use XB kinetics, time-resolved PL and NIR-TA cation signals (not measured there) to clearly establish the picture of hole transfer mediated TET. The comparison between the NC-TCA and NC-NCA

data thus leads us to propose the unified model for TET in these NC/PAH systems. As acknowledged by the reviewer, such a model should be very important for the field of TET between inorganic NCs and organic molecules.

But is a bit puzzling in that it shows different results as only charge separation was seen in the Chem. Sci. work with no subsequent triplet formation. The discrepancy between these two studies could reflect the difference in the bandgap of the perovskite nanocrystals employed in each. However, I don't think this should be the case as a wider bandgap perovskite nanocrystal was used in the Chem. Sci. work, which should provide a larger driving force for triplet energy transfer.

Response: We thank the reviewer very much for the critical insight. Indeed, making a connection and discuss the differences between the current work and the previous Chem. Sci. should help us greatly improve the quality of the paper.

As we mentioned above, the charge recombination process we reported in the Chem. Sci. paper is essentially electron transfer from NC^- to TCA^+ to form TCA triplets. The difference between the data in the Chem. Sci. paper and here is then only the rate of the electron transfer process. Electron transfer occurs in ~ 1.9 ns here, whereas it was extremely slow (~ 5.1 μs) in the previous work.

We would like to point out that the bandgap of the $\text{CsPbCl}_x\text{Br}_{3-x}$ NCs ($\text{Cl}:\text{Br} = 1/8$) used in the Chem. Sci. paper (absorption peak at ~ 490 nm) is not wider but narrower than the NCs used here (absorption peak at ~ 460 nm). But the difference in the electron transfer rate is indeed related to the sizes of the NCs (quantum confinement effect).

In order to illustrate this point, we have prepared CsPbBr_3 NCs of various sizes from ~ 2.6 to 8.5 nm, corresponding to first-exciton absorption peaks from ~ 440 to 500 nm. We measured the TA and time-resolved PL kinetics of NC-TCA complexes with different NC sizes and the data are summarized in Fig. R6. All the TA XB kinetics show fast decay followed by slower decay, corresponding to the processes of hole transfer and electron transfer, respectively. In contrast, the PL kinetics show only fast decay within ~ 1 ns, consistent with the hole transfer process. Obvious from Fig. R6 is that both hole transfer and electron transfer rates show a size dependence. The hole transfer rate slows down as the size increases (the data above the gray dashed line in Fig. R6a and the data in Fig. R6b), as does the electron transfer rate (the data below the gray dashed line in Fig. R6a). According to previous studies on charge transfer from NCs, NC size affects charge transfer rate for two reasons. First, the band edge energy levels shifts to higher energy with decreasing NC size and hence the charge transfer driving force increases; second, the charge transfer electronic coupling term increases with decreasing NC size because small NCs have more wavefunction leakage on their surfaces.

A quantitative analysis of the size-dependent energetics and electronic coupling terms and their correlation to the hole transfer and electron transfer rates are beyond the scope of this work. But the trend shown in Fig. R6a is consistent with a NC size effect. The slow electron transfer observed for the largest-size, ~ 8.5 -nm CsPbBr_3 NCs is approaching that of the ~ 10 nm $\text{CsPbCl}_x\text{Br}_{3-x}$ NCs used in the Chem. Sci. paper.

Fig. R6. Normalized XB (a) and TR-PL (b) kinetics of a series of NC-TCA complexes with different NC sizes. The sizes of these NCs are tuned from ~ 2.6 to 8.5 nm, corresponding to first-exciton absorption peaks from 440 to 500 nm which are used to label the NCs in the figure. The wine-colored line in (a) is from ref which used a sample of ~ 10 -nm $\text{CsPbCl}_x\text{Br}_{3-x}$ NCs. The TA data above and below the gray dashed line in (a) mostly reflect hole and electron transfer, respectively, whereas the TR-PL data in (b) measure hole transfer only. Both hole and electron transfer slow down with increasing NC sizes.

Revision: In the paragraph where we fit the hole and electron transfer time constants for NC-TCA complexes, we add the following discussion to clarify the above-mentioned point:

“It is also interesting to note that while the lifetime of the $\text{NC}^- \text{TCA}^+$ charge separated states is only ~ 1.9 ns, it is as long as ~ 5.1 μs in our previous report using $\text{CsPbCl}_x\text{Br}_{3-x}$ NCs.⁴³ This is likely a NC size effect. As shown in Supplementary Fig. 9, both hole transfer and electron transfer slow down substantially with increasing CsPbBr_3 NC sizes, because both the energetics and electronic coupling terms involved in charge transfer from quantum-confined NCs depend sensitively on NC sizes.⁴⁸ For

8.5-nm CsPbBr₃ NCs, the electron transfer time (i.e., charge separated state lifetime) starts to approach that of the 10-nm CsPbCl_xBr_{3-x} NCs studied previously.⁴³,

In the SI, we add Fig. R6 as a new Supplementary Fig. 9.

While the experimental work carries less impact due to its appearance elsewhere, the theoretical approach to compute driving forces for charge separation and triplet energy transfer is new and offers a potential means to rectify the different dynamics the authors observe in their two systems. However, this model needs some additional vetting before publication as its accuracy in computing excitonic energy levels are off relative to experiment to a large enough level that I am not sure its mechanistic predictions can be trusted for the systems the authors investigate. In particular, I suspect that it would predict that the tetracene system investigated in the authors' prior Chem. Sci. work would also go on to form triplet excitons. While I appreciate the authors' approach and think it is well thought out, one would like to see it vetted using experimental values for redox properties measured by electrochemical, spectroscopic, and photoelectron methods rather than using purely computed values.

Overall, I feel the experimental data alone does not merit publication in Nature Communications. However, if packaged with a more convincing theoretical model, vetted by experimental data rather than computed energy levels, which is shown to not only explain data in this publication but also work the authors and other groups have published in looking at perovskite to acene triplet energy transfer, then I think this manuscript could carry the import needed for publication in Nature Communications.

Response: We thank the reviewer for this insightful comment. The major concern of the reviewer, as I can see, is the accuracy of the energy levels used in the calculation, which determined how convincing our proposed TET models can be. **Per his/her suggestion, we have measured the energy levels in the molecules and NCs using cyclic voltammogram (CV). As will be elaborated below, these numbers are indeed slightly different from those we used in our previous Fig. 1, but the general conclusions remain valid.**

Below, I identify additional specific issues for the authors to address while revising their manuscript.

1) A key issue I have with this manuscript is the vetting of the energy level diagram shown in figure 1c. This picture serves as the basis for the authors' conclusion that tetracene is capable of accepting a hole from a perovskite nanocrystal but naphthalene is not. However, I question the accuracy of the energy levels pictured. For example, the energy levels shown for the perovskite imply a bandgap of ~3 eV, but the optical spectra shown in figure 1d right below show a band edge exciton transition at ~2.7 eV.

Response: We thank the reviewer for pointing out this difference which indeed seems confusing at first glance. But we would like to point out that the energy levels shown in Fig. 1c are for "single-particle" states. For NCs, it means only the electron or hole

is in the NCs. In optical excitation, however, an electron-hole pair is created which is bound by the coulomb energy. Thus, the difference between the single-particle electron-hole gap (~ 3 eV) and the optical gap (~ 2.7 eV) is due to the electron-hole binding energy. Our calculations in Supplementary Note 1 have explicitly included these coulomb energy terms.

These levels also appear to be shifted down with respect to vacuum by ~ 0.3 eV relative to values reported by the authors in their Chem. Sci. paper.

Response: The shift is reasonable considering the strong quantum confinement effect in the NCs used here (3.8 nm size) compared to the bulk-like NCs used in the Chem. Sci. paper (~ 10 nm).

These bandshifts are troubling given that the authors point to barrier of only ~ 30 meV (on the order of $k_B T$) to argue as to why hole transfer between the perovskite valence band and naphthalene does not occur.

Response: The driving forces for charge transfer reactions are not simply determined by the “single-particle” energy alignments shown in Fig. 1c. Rather, we need to account for various Coulombic binding and charging energies involved in CT. For example, when examining hole transfer from NCs to NCA, we should consider the energy penalty associated with breaking the electron-hole pair in NCs and putting extra charges into NCs and NCA as well as the energy compensation from electron-hole binding in the charge separated states ($NC^- - NCA^+$). These would lead to an extra barrier of 0.4 eV to the hole transfer process, thus inhibiting hole transfer. Our calculations in Supplementary Note 1 have explained these details.

Energy levels computed for tetracene appear to be equally off, which is computed to have a band gap of 2.3 eV, but appears to have one closer to 2.6 eV looking at optical spectra in figure 1d.

Response: The gap of organic molecules is somewhat confusing in the literature because some used absorption peak whereas others used absorption onset. A more appropriate approach that is generally accepted in the community is to use the crossing point between the normalized absorption and luminescence spectra to determine the gap (*Nat. Mater.* **2018**, *17*, 119.) because this crossing point corresponds to the energy of the transition from the zeroth vibrational ground state to the zeroth vibrational first excited state. As shown in Fig. R7, the crossing point is at ~ 517 nm or 2.4 eV for TCA.

Fig. R7. Normalized absorption and PL spectra of TCA used to determine its band gap.

Do the authors have any means available to experimentally vet the energy levels shown in figure 1c such as CV or photoelectron spectroscopy? Without such data, I find difficult to believe that the redox properties of the two materials alone is enough to drive the difference in dynamics they observe between their two systems. Providing such experimental vetting is key as the energy level diagram in figure 1c forms the crux for their conclusions.

Response: We thank the reviewer very much for the suggestion of experimentally measure the energy levels instead of simply citing or calculating the numbers. Per this suggestion, we have performed CV measurements for NCA and TCA, as well as for CsPbBr₃ NCs. The results are shown in Figs R8-R10. For every measurement, we also measured the Ferrocene/Ferrocenium (Fc/Fc⁺) standard under exactly the same conditions as the sample, in order to ensure the accuracy of the results.

The redox couples labeled in Fig. R8b can be assigned to the HOMO or oxidation potential of NCA. The value of the oxidation potential energy with respect to vacuum can be calculated as: $E_{\text{ox}} = -[V_{\text{ox}} - V(\text{Fc}/\text{Fc}^+) + 4.8] \text{eV} = -[(1.5 + 1.3)/2 - (0.3 + 0.6)/2 + 4.8] = -5.8 \text{ eV}$. In this equation, the redox potentials of the molecules were calculated as the mean values of the cathodic and anodic peaks according to the convention. 4.8 is the potential of Fc/Fc⁺ with respect to vacuum. Because the LUMO or reduction potential of NCA was not obtained in the used electrochemical window. We calculate it using: $E_{\text{red}} = E_{\text{ox}} + E_{\text{g}} = -5.8 + 3.8 = -2 \text{ eV}$, with E_{g} being the optical gap of NCA that can be determined from the the crossing point between the normalized absorption and luminescence spectra in Fig. R8c.

Fig. R8. CV curves of Ferrocene/Ferrocenium (Fc/Fc^+) standard (a) and NCA (b) measured at RT. The CVs were performed in acetonitrile at a scan speed of 100 mV s^{-1} . The redox couples labeled in (b) can be assigned to the HOMO or oxidation potential of NCA. The value of the oxidation potential energy with respect to vacuum can be calculated as: $E_{\text{ox}} = -[V_{\text{ox}} - V(\text{Fc}/\text{Fc}^+) + 4.8] \text{ eV} = -[(1.5 + 1.3)/2 - (0.3 + 0.6)/2 + 4.8] = -5.8 \text{ eV}$. In this equation, the redox potentials of the molecules were calculated as the mean values of the cathodic and anodic peaks. 4.8 is the potential of Fc/Fc^+ with respect to vacuum. Because the LUMO or reduction potential of NCA was not obtained in the used electrochemical window. We calculate it using: $E_{\text{red}} = E_{\text{ox}} + E_{\text{g}}$, with E_{g} being the optical gaps of the molecules. E_{g} of NCA can be determined from the the crossing point between the normalized absorption and luminescence spectra in (c). E_{red} of TCA can thus be calculated as: $E_{\text{red}} = E_{\text{ox}} + E_{\text{g}} = -5.8 + 3.8 = -2.0 \text{ eV}$.

Similarly, E_{ox} of TCA (Fig. R9) can be calculated as: $E_{\text{ox}} = -[V_{\text{ox}} - V(\text{Fc}/\text{Fc}^+) + 4.8] \text{ eV} = -[(1.37 + 0.63)/2 - (1.29 + 0.01)/2 + 4.8] = -5.1 \text{ eV}$. E_{red} of TCA can be calculated as: $E_{\text{red}} = E_{\text{ox}} + E_{\text{g}} = -5.1 + 2.4 = -2.7 \text{ eV}$,

Fig. R9. CV curves of Ferrocene/Ferrocenium (Fc/Fc^+) standard (a) and TCA (b) measured at RT. The CVs were performed in dichloromethane at a scan speed of 100 mV s^{-1} . The redox couples labeled in (b) can be assigned to the HOMO or oxidation potential of TCA. (c) Normalized absorption and luminescence spectra of TCA used to determine its E_{g} .

For CsPbBr_3 NCs, both electron and hole energy levels can be determined from the CV curve (Fig. R10b). However, there are additional peaks (labeled by blue dashed lines) that cannot assigned to the band edge energy levels because the energy

gaps calculated from these peaks are not consistent with experimental gaps. These peaks were also reported in a previous CV study on perovskite NCs and were assigned to additional species in the solution such as the unreacted Pb-oleate (*ACS Energy Letters* **2016**, *1*, 665-671). Also note that E_e and E_h are irreversible peaks so that their values are determined from single peaks instead of mean values of cathodic and anodic peaks, which are consistent with previous CV studies on NCs (*ACS Energy Letters* **2016**, *1*, 665-671; *J. Am. Chem. Soc.* **2001**, *123*, 8860–8861. *J. Chem. Phys.* **2003**, *119*, 2333–2337). As such,

$$E_e = -[V_e - V(\text{Fc}/\text{Fc}^+) + 4.8] \text{eV} = -[-1.4 - (0.4 + 0.8)/2 + 4.8] = -2.8 \text{ eV};$$

$$E_h = -[V_h - V(\text{Fc}/\text{Fc}^+) + 4.8] \text{eV} = -[1.5 - (0.4 + 0.8)/2 + 4.8] = -5.7 \text{ eV}.$$

The difference between E_e and E_h is 2.9 eV, which is higher than the optical gap of the NCs (~2.7 eV) because of a strong electron-hole coulomb binding energy in NCs.

Fig. R10. CV curves of Ferrocene/Ferrocenium (Fc/Fc^+) standard (a) and CsPbBr_3 NCs (b) measured at RT. The CVs were performed in acetonitrile/toluene mixture (1:4 v/v) at a scan speed of 100 mV s^{-1} . V_e and V_h are labeled in (b) and the extra peaks labeled by blue dashed lines were likely due to additional species in the solution such as the unreacted Pb-oleate.

Revision: We add the following two paragraphs describing measurements of energy levels in these materials using CV:

“We used two carboxyl-functionalized PAH molecules, 1-naphthalene carboxylic acid (NCA) and 5-tetracene carboxylic acid (TCA), as the triplet acceptors (Fig. 1b). The redox potentials of unsubstituted naphthalene and tetracene have been reported,¹ but, considering the potential effect of the carboxyl group, we measured the redox potentials of NCA and TCA molecules using cyclic voltammetry (CV); see Methods for details. For both NCA and TCA, only their oxidation potential energies (E_{ox}) could be determined from CV in the used electrochemical window (Supplementary Figs. 2 and 3) and their reduction potential energies (E_{red}) were approximated as: ^{3,34,35} $E_{\text{red}} = E_{\text{ox}} + E_g$, with E_g being the optical gaps of the molecules. Similarly, the reduction potential energies to form molecular triplets ($E_{\text{red,T}}$) were approximated as: ^{3,34,35} $E_{\text{red,T}} = E_{\text{ox}} + E_T$, with E_T being the triplet energies. For CsPbBr_3 NCs, both the lowest electron and hole energies (E_e and E_h , respectively) were determined from CV (Supplementary Fig. 4), with some additional intragap features that were tentatively

assigned to lead oleate species in the solution.³⁶

The determined energy level alignments in the NC-PAH systems are summarized in Fig. 1c. The energy levels of NCA and TCA are indeed shifted by 100s of meV compared to those reported for unsubstituted naphthalene and tetracene.¹ The energy difference between E_e and E_h determined for the NCs is ~ 2.9 eV, which is consistent with the optical gap of 2.7 eV (460 nm) after accounting for an electron-hole binding energy of ~ 0.2 eV in 3.8-nm CsPbBr₃ NCs (Supplementary Note 1). This consistency also suggests that while the absolute values of the energy levels in Fig. 1c are only for reference due to many well-known complications associated with these measurements, the energy level alignments (i.e., relative values) should be reliable.”

These results are summarized in a revised Fig. 1c:

Figure 1. Design of the NC-PAH systems for TET study. (c) Schematic energy level alignment between NCs and NCA and TCA determined from cyclic voltammogram. E_e and E_h are the lowest electron and hole energy levels in the conduction and valence bands, respectively. The difference between them is ~ 2.9 eV, which is higher than the optical gap of the NCs (~ 2.7 eV) because of a strong electron-hole coulomb binding energy in NCs. E_{ox} , E_{red} , and $E_{red,T}$ are the ground state oxidation potential energy, reduction potential energy and triplet state reduction potential energy, respectively, of the PAH molecules. E_T is the triplet energy.”

Realizing that charge transfer driving force calculations are somewhat confusing if we simply look at the numbers reported in Fig. 1c, we add the following paragraph describing how charge transfer driving forces are calculated and a new Fig. 2 to illustrate the energy of various charge transfer states with respect to that of the initial photoexcited states:

“The driving forces for CT reactions ($-\Delta G_{CT}$), however, are not simply determined by the “single-particle” energy alignments shown in Fig. 1c. Rather, we need to account for various Coulombic binding and charging energies involved in CT. For example, when examining hole transfer from NCs to NCA, we should consider the energy penalty associated with breaking the electron-hole pair in NCs and putting extra charges into NCs and NCA as well as the energy compensation from electron-hole binding in the charge separated states (NC⁻-NCA⁺). Detailed for these calculations are provided in Supplementary Note 1. In Fig. 2, we plot the calculated

energies of various CT states ($\text{NC}^- \text{-NCA}^+$, $\text{NC}^+ \text{-NCA}^-$, $\text{NC}^- \text{-TCA}^+$ and $\text{NC}^+ \text{-TCA}^-$) relative to that of photoexcited NCs (NC^*) in a Marcus-type reaction coordinate diagram; see also Supplementary Table 1. According to the diagram, electron transfer from photoexcited NCs to both NCA and TCA ground states is energetically unfavourable. On the other hand, hole transfer from photoexcited NCs to NCA and TCA ground states should be energetically disallowed and favoured, respectively.

Figure 2. CT energetics. The parabolas represent the reactant state ($\text{NC}^* \text{-PAH}$) and the various CT product states drawn on a continuous reaction coordinate mainly contributed by the surrounding medium. The lowest energies of the CT states with respect to that of the $\text{NC}^* \text{-PAH}$ state are indicated.”

In addition, in the SI, Figs. R8-R10 were added as new Supplementary Figs. 2-4.

2) It is clear the authors have two different kinetic models in mind for describing both the dynamics of their tetracene:perovskite system (perovskite exciton \square charge separated state \square tetracene triplet) and tetracene:naphthalene system (perovskite exciton \square naphthalene triplet). I am a bit puzzled then as to why the authors fit their kinetic data to a series of single exponential or biexponential functions at specific probe wavelengths rather than using a global analysis package to fit their full transient spectra data sets to the solution of a coupled set of kinetic equations. This fit could be applied simultaneously to both their visible and near-infrared spectral data and would provide a more natural interpretation of the rate constants recovered from their model.

Response: We thank the reviewer very much for this suggestion. Indeed, ideally, a global analysis package can be used to simultaneously capture the coupled kinetics of the species (NC excited states, PAH cations and triplets). However, in practice, we find that **because the extinction coefficients of NCs are orders of magnitudes stronger than those of the PAH states**, it is very difficult to use such a global fit to correctly reproduce the kinetics of the PAH states as they are overwhelmed by NC signals. This is different from the case of, e.g., studying singlet fission in PAH molecules where all the involved states are molecular states with comparable

extinction coefficients. Thus, for the NC-PAH hybrid systems, it is more practical to find the characteristic wavelengths of molecular species where NC absorption is negligible and to plot the kinetics at these specific wavelengths to represent the kinetics of these molecular species. Such an approach is commonly used in studies of NC-PAH hybrid systems, such as the *Science* paper by Castellano et al. (Direct observation of triplet energy transfer from semiconductor nanocrystals. *Science* **2016**, *351*, 369-372.)

3) Have the authors characterized the average number of tetracene/naphthalene molecules that bind to each perovskite nanocrystal? If these values are known, then presumably the rate constants the authors extract from their data could be scaled to account for the fact that multiple molecules bind to each nanocrystal, thereby accelerating rates measured for energy transfer. Presumably, the reported rates for triplet/charge transfer are enhanced to a large degree by the number of acceptors.

Response: We thank the reviewer for this suggestion. The average number of PAH molecules per NC was reported in the Method section of “**Preparation of NC-PAH complexes**”, which is about 60. Indeed, the charge transfer and triplet transfer rates are strongly enhanced by the number of acceptors. This was already acknowledged in our description of ultrafast hole transfer in NC-TCA: “Specifically, hole transfer is accelerated by the availability of multiple acceptors as the rate should scale approximately with number of adsorbed acceptors⁴⁸”

4) Is there any evidence for modification of the absorption features of tetracene/naphthalene upon binding to the perovskite? Looking at the absorption onset for tetracene in particular (figure 1d), there appears to be a weak shoulder that extends out past 500 nm that could indicate some sign of aggregation of tetracene monomers on the surface.

Response: We thank the reviewer for this good suggestion. We have carefully compared the absorption spectra of NCA and TCA on NC surfaces obtained from spectral differences between complexes and free NCs to the absorption spectra of NCA and TCA dissolved in toluene. We find that the absorption spectra of NCA on NC surfaces and in toluene solution are very similar. But, interestingly, the spectral features of TCA on NC surfaces **are narrower than** those of TCA in toluene solution. This suggests that TCA molecules likely aggregate in solution, which is expected for extended π -systems and is consistent with previous reports on singlet fission observed for tetracene derivatives in solution. In contrast, TCA molecules are well-dispersed in the ligand shell on NC surfaces, resulting in narrow line-shapes.

Revision: We add a new Fig. 3c to describe the absorption changes of molecules upon adsorbing on NC surfaces:

“**Figure 3. Optical properties of the NC-PAH systems.** (c) Absorption spectra of NCA and TCA on NC surfaces (blue and orange solid lines, respectively) obtained from spectral differences in (b) and absorption spectra of NCA and TCA dissolved in toluene (blue and orange dashed lines, respectively).”

5) The authors estimate that holes contribute to 25% of the XB photobleach of the perovskite nanocrystals, but I find the data in supplementary figure 2 from which this conclusion is drawn very confusing. What I would expect to see in the XB dynamics shown in panel b would be multiexponential decay kinetics, with an initial rate corresponding first to electron transfer to rhodamine B followed by a separate slower decay due to charge recombination. The electron transfer step should be reflected by a bleaching of rhodamine B followed by a decay of the rhodamine bleach as charge recombination occurs. While it does appear that the growth of the rhodamine bleach tracks the initial portion of the perovskite bleach decay, after ~ 1 ns the rhodamine bleach plateaus while the perovskite bleach continues to decay. There also doesn't appear to be multiple decay timescales that can be readily seen in the perovskite bleach recovery. This makes me question if the amplitude of the decay can fully be ascribed to only forward and back charge transfer to rhodamine. If another process impacts the perovskite bleach recovery, this would invalidate the author's assignment of 25% of the XB photobleach to the hole.

Response: We thank the reviewer for very much for this very insightful comment. Yes, the data presented in our previous Supplementary Fig. 2 are oversimplified and seem contradictory to the model of electron transfer followed by charge recombination. In fact, these data are likely consistent with a **physical picture of electron transfer mediated triplet energy transfer from NCs to RhB**, which we didn't elaborate in our previous version because we thought it would be too much involved. But we do agree with the reviewer that without clarifying these observations the validity of the electron and hole contribution assignments is compromised.

Fig. R11. (a) TA spectra of NC-Rhodamine B (NC-RhB) complexes probed at indicated delays following the excitation by a 400 nm pulse which selectively excites NCs. (b) Comparison of XB (scaled by a factor of 5) and RhB kinetics. The formation of the RhB signal is complementary to the decay of the XB signal of NCs within ~ 10 ns; after that, the XB decays whereas the RhB signal is longer-lived.

In Fig. R11, we have added ns-TA data to show XB and RhB bleach kinetics at later delays. We find that XB shows $\sim 75\%$ decay within ~ 10 ns and the RhB bleach gradually reaches its maximum. This process is consistent with electron transfer from photoexcited NCs to RhB. The confusing part is that, at longer delays, the rest part of the XB signal decays in $1 \mu\text{s}$ whereas the RhB bleach does not. In fact, the RhB bleach is extremely long-lived, showing a lifetime of $\sim 50 \mu\text{s}$. Considering the short lifetime (a few ns) of RhB singlet excited state, the long-lived RhB bleach is most likely due to RhB triplets. Therefore, hole transfer from NC^+ to RhB^+ to form a spin triplet state. Although the charge recombination process does not simultaneously annihilate RhB bleach and NC hole signals, we can choose the point at which the RhB bleach reaches its maximum as the finishing point of electron transfer process. The XB amplitudes before and after this time point can then be assigned to electron and hole contributions, respectively, which yield an electron contribution of 75% and hole

contribution of 25%.

However, because the mechanism described above is beyond the scope of this work and possibly more experiments will be required to further confirm it, we decide to remove this part from the paper. This would not affect the assignment of electron and hole contributions in our work, because the NC-TCA experiment itself is a very convincing assignment already, particularly after that we have confirmed the NIR spectral features as TCA cation signals (see our response to reviewer 1; Figs R2 and R3).

Revision: We have removed Supplementary Fig. 2 from the SI, and the statement in the main text has been changed to:

“The contributions of the electron and the hole to the XB are determined to be ~75% and 25%, respectively, based on the NC-TCA experiment to be described below.”

As a minor followup question, why does the XB photobleach grow for the first ps following photoexcitation? Is this tied to carrier cooling within the perovskite nanocrystal?

Response: Yes, this is a typical signature of hot carrier cooling in NCs. The NCs were excited at 400 nm and it took them ~1 ps to fully relax to the band edge states at ~460 nm. This observation is consistent with TA studies of perovskite NCs reported in the literature.

6) For the tetracene:perovskite system, I would expect hole transfer to tetracene to lead to a photobleach of tetracene absorption features in the visible spectral range, but evidence of tetracene photobleaching is not readily apparent looking at data shown in figure 3c. Presumably this reflects the difference in extinction between the perovskite and tetracene triplet acceptors which could make the tetracene bleach difficult to observe. However, this extinction ratio is known and the authors should be able to compute the estimated strength of the tetracene photobleach if full each perovskite photoexcitation produced a tetracene cation. I suspect this should give some observable features in the author’s transient spectra that could be associated with photoexcited tetracene. If the authors find the tetracene photobleach is estimated to be larger than that observed experimentally, does this perhaps provide some evidence that the hole may not be fully displaced onto the tetracene core?

Response: We thank the reviewer very much for this comment. Indeed, hole transfer should in principle bleach the absorption of TCA. The difficulty of observing the bleach is, as pointed out by the reviewer, due to a ~100-fold difference in the extinction coefficients of NCs and ground-state TCA. This is illustrated by a careful comparison of the TA spectra of NCs and NC-TCA at delay times of, e.g., 25 ps and 50 ps when hole transfer has mostly finished. In Fig. R12a, we show that after scaled to the same amplitude at the XB maximum, the TA spectra of NC and NC-TCA at 25 ps are indeed slightly different. Taking a difference between them leads to a derivative-like spectrum. This derivative spectrum should contain a contribution from the TCA bleach, but it should also be contributed by charge-separation induced

stark-effect-like signals that have been extensively reported for NC-molecule charge separated states. As such, it is unpractical to extract the accurate contribution by the TCA bleach and to quantify the hole transfer yield using this bleach amplitude. Fig. R12b shows the spectral comparison at 50 ps giving the same result.

Fig. R12. (a) TA spectra of NCs (black) and NC-TCA complexes (red) at a delay of 25 ps. The latter can be scaled by a factor (blue) to match the XB maximum of the former. The difference between the scaled NC-TCA spectrum and the NC spectrum yields a derivative-like spectrum (green). (b) The same plot as (a) but for a time delay of 50 ps.

Although we cannot quantify the hole transfer yield directly from the TCA bleach suggested by the reviewer, we can use the TCA triplet signal to quantify the overall triplet formation yield. This will be illustrated below in our response to the reviewer’s comment (8). The quantified TCA triplet formation yield is 94.8%, very close to the hole transfer yield we calculated from the hole transfer time constant. Thus, we think that hole transfer yield indeed reaches 98.6%, otherwise we cannot attain a triplet formation yield as high as ~95%. We hope that reviewer can find this rationale useful.

Revision: In the section where we describe hole transfer, we add the following content to explain why the TCA bleach is not observed:

“During the hole transfer process, we should expect the formation of a ground state bleach (GSB) feature of TCA. This observation, however, is hindered by a spectral overlap between the GSB of TCA and XB of NCs and by a ~100-fold difference between the extinction coefficients of NCs ($\sim 878000 \text{ M}^{-1}\text{cm}^{-1}$ at 460 nm) and ground-state TCA ($\sim 7400 \text{ M}^{-1}\text{cm}^{-1}$ at 482 nm).”

7) Looking at the optical absorption and emission spectra reported for the perovskite:tetracene system (figures 1d & 1e), I’m a bit surprised that FRET energy transfer from the perovskite to tetracene is not considered as the perovskite emission perfectly overlaps the tetracene absorption. I realize this pathway was previously considered by the authors in their Chem. Sci. paper earlier this year and deemed too slow to be important, but that manuscript reported data on a perovskite with a

narrower bandgap that should form a worse FRET pair. Is it possible that the charge transfer step the authors observe first involves formation of a singlet exciton on tetracene followed by charge transfer back to the perovskite? How would the authors rule out such a pathway?

Response: We thank the reviewer very much for this insightful comment. However, as we pointed out above, bandgap of the CsPbCl_xBr_{3-x} NCs (Cl:Br = 1/8) used in the Chem. Sci. paper (absorption peak at ~490 nm) is not wider but narrower than the NCs used here (absorption peak at ~460 nm). In Fig. R13, we plot the molar extinction coefficient of TCA and the PL spectra of the NCs used here and used in the Chem. Sci. paper; the PL spectra are rescaled to match the absorption spectrum at their maxima. Using this figure, we calculate that the FRET overlap integrals are 3.08×10^{14} and $3.13 \times 10^{14} \text{ M}^{-1}\text{cm}^{-1}\text{nm}^4$, respectively, for the NCs used here and used in the Chem. Sci. paper. Thus, the overlap integrals are very similar, instead of being much worse for the Chem. Sci. paper as suggested by the reviewer.

Fig. R13. The molar extinction coefficient of TCA (black) and the PL spectra of the NCs used here (blue) and used in the Chem. Sci. paper (red); the PL spectra are rescaled to match the absorption spectrum at their maxima.

Careful inspection of the NC-TCA spectroscopy data also shows that FRET is not a applicable model. The TA kinetics of NC-TCA complexes probed at the XB shows an ultrafast decay within 50 ps with a relative amplitude of ~25%, while on a similar timescale the TR-PL exhibits complete decay. **If it was indeed due to FRET, we should expect complete decay of both the XB and the TR-PL rather than partial decay of the XB and complete decay of the TR-PL, as energy transfer should simultaneously annihilate both the electron and the hole.** Also, as suggested by the reviewer, FRET from NCs to TCA might be followed by electron transfer back from TCA to NCs. If this was the case, we should observe complete decay of the XB, followed by gradual formation again of the XB due to this back electron transfer. Our experimental data clearly do not agree such an expectation.

On the basis of the above-mentioned reasons, we believe that despite of a significant spectral overlap, FRET plays a negligible role in NC-TCA.

Revision: In the section where we describe the optical properties of the NC-PAH systems, we add the following content to acknowledge that FRET is in principle energetically allowed:

“In addition, because of a strong overlap between the PL spectrum of NCs and the absorption spectrum of TCA, Förster resonant energy transfer (FRET) from NCs to TCA is also allowed. Later we will use time-resolved spectroscopy to clarify the role of these quenching pathways.”

In the section where we describe the NC-TCA spectroscopy data, we add the following content to rationalize how we can rule out FRET:

“The TA kinetics of NC-TCA complexes probed at the XB shows an ultrafast decay within 50 ps with a relative amplitude of ~25%, while on a similar timescale the TR-PL exhibits complete decay (Fig. 5b). On the basis of the energetics analysis for NC-TCA above, these observations are most consistent with ultrafast hole transfer from NCs to TCA to form the NC⁻-TCA⁺ charge separated state and, accordingly, the electron and hole contributions to the XB are 75% and 25%, respectively. The other two energetically allowed pathways, FRET and direct TET, should lead to complete decay of both the XB and the TR-PL rather than partial decay of the XB and complete decay of the TR-PL, as energy transfer should simultaneously annihilate both the electron and the hole. For the FRET pathway, in particular, we should also expect complete decay of the XB followed by gradual formation again of the XB due to energetically allowed back electron transfer from TCA to NCs; such a peculiar kinetic behaviour is clearly not consistent with our experimental observations. The dominance of hole transfer over FRET and direct TET is consistent with previous reports on related systems.^{43,44,}”

8) Estimates for the triplet extinction of both tetracene and naphthalene are available as well as the tetracene cation, and I would encourage the authors to estimate the yields of producing each of these species from their transient data. Do all excitations eventually go on to form triplets? If not, where are they being lost? Does charge recombination to the ground state in the tetracene system play a major competing role? A naïve conclusion based on the weak triplet induced absorption relative to the perovskite features shown in figure 3 would lead one to conclude that triplet energy transfer is somewhat inefficient in both systems, which would hamper the utility of these materials for applications.

Response: We thank the reviewer very much for this useful suggestion. Previously, we calculate the TET yields, which are the triplet formation yields, based on charge and energy transfer time constants. Indeed, the triplet formation yields can also be estimated based on transient signal amplitudes of NCs and molecular triplets and their reported extinction coefficients. In the SI, we add a new Supplementary Note 5 to illustrate how the triplet formation yields in NC-TCA and NC-NCA are estimated. According to these estimations, the triplet formation yields in NC-TCA and NC-NCA are 94.8% and 64.3%, respectively, which are in excellent agreement with the TET yields calculated using time constants (98.6% and 65.0%, respectively). This agreement validates the accuracy of both types of calculations.

That the triplet absorption signals are very weak on the TA spectra is, again, simply a result of ~20-fold difference between the extinction coefficients of molecular triplets and NCs. It is not an indication of low TET yields in these systems.

Revision: At the end of the NC-TCA section, we add the following content:

“In this case, the overall TET yield is determined by the hole transfer yield. Indeed, the triplet formation yield, estimated using the maximum signal amplitudes of $^3\text{TCA}^*$ absorption and NC bleach and the extinction coefficients reported for TCA triplets¹ and NCs⁴⁹ (Supplementary Note 5), is ~94.8%, similar to the TET yield determined from kinetic parameters. We note that the very weak $^3\text{TCA}^*$ signal observed in Fig. 5b is simply a result of the 20-fold difference between the extinction coefficients of $^3\text{TCA}^*$ and NCs.”

Similarly, in the NC-NCA section, we add the following content:

“Fitting the kinetics in Fig. 5d revealed an averaged TET time of 2.1 ± 0.1 ns and a TET yield of ~65.0% (Supplementary Note 2 and Table 4). The calculated TET yield is consistent with the NCA triplet formation yield (~64.3%) estimated from TA signal amplitudes (Supplementary Note 5).”

In the SI, we add a new Supplementary Note 3:

“Supplementary Note 3. Estimation of triplet formation yields

The CsPbBr₃ NC-to-PAH triplet formation yields were determined by using ultrafast TA data. For the CsPbBr₃ control spectra, the ΔA at the wavelength of exciton bleach (XB) and the corresponding ground state molar extinction coefficient (ϵ_1) for the CsPbBr₃ NCs at the XB peak ($\sim 615000 \text{ M}^{-1}\text{cm}^{-1}$)⁸ were used to calculate the concentration of NC excited states (NC^*) generated. For the NC-PAH samples, the experiments were performed under identical experimental conditions. The concentration of $^3\text{PAH}^*$ formed was determined using the maximum ΔA at ~425 nm for NCA (465 nm for TCA) and its corresponding triplet molar extinction coefficient ϵ_2 .⁹ Thus, the comparison of the concentration of CsPbBr₃ NC^* generated by the laser pulse in absence of PAH acceptor and the concentration of $^3\text{PAH}^*$ generated from TET permitted the determination of the triplet formation yield:

$$\Phi_{TET} = \frac{[{}^3\text{PAH}^*]}{[\text{NC}^*]} = \frac{\Delta A_{{}^3\text{PAH}^*} / \epsilon_2}{\Delta A_{\text{NC}^*} / (\epsilon_1 / 2)} = \frac{\Delta A_{{}^3\text{PAH}^*} \epsilon_1}{2 \Delta A_{\text{NC}^*} \epsilon_2} \quad (\text{S14}).$$

Note that the factor of 2 accounts for the two-fold spin-degeneracy of the band edge state in CsPbBr₃ NCs, that is, one exciton in a NC only bleaches half of its absorption. The calculated TCA and NCA triplet formation yields are listed as follows.

$$\Phi_{TET(\text{NC-TCA})} = \frac{\Delta A_{{}^3\text{TCA}^*} \epsilon_1}{2 \Delta A_{\text{NC}^*} \epsilon_2} = \frac{1.25 * 615000}{2 * 13 * 31200} = 94.8\% \quad (\text{S15}),$$

$$\Phi_{TET(\text{NC-NCA})} = \frac{\Delta A_{{}^3\text{NCA}^*} \epsilon_1}{2 \Delta A_{\text{NC}^*} \epsilon_2} = \frac{0.4 \times 615000}{2 \times 14.5 \times 13200} = 64.3\% \quad (\text{S16}).$$

”

9) On p. 11, the authors make reference to “through-configuration” pathways contributing to the matrix element for triplet energy transfer from a perovskite nanocrystal to naphthalene. This type of contribution has been referred to in the literature by a few different names, including as a “superexchange” contribution, “charge resonance” contribution, and “virtual charge transfer” contribution. It would be worth making use of some of these other terms for clarity as I believe “through-configuration” is not often used.

Response: We thank the reviewer for this suggestion. We have now rephrased our words throughout the manuscript to only use the term “through-configuration” when we are citing the original paper by Harcourt et al. In other places, we use the more widely-used term of “virtual charge transfer”.

Revision: We have rephrased our words throughout the manuscript.

10) The value reported for the averaged electron transfer time constant (1878 +/- 74 ps) seems very precise given the error and accuracy of other values reported in the manuscript.

Response: We thank the reviewer for this comment. This number is directly calculated from fitting parameters. But we agree with the reviewer that it is not that meaningful to keep so many significant numbers in light of the error and accuracy of other values reported in the manuscript.

Revision: We change it to 1.9 ± 0.1 ns.

11) Minor point, on p. 5 I would state “ground state NCA and TCA” rather than “NCA and TCA ground states”.

Response: We thank the reviewer for this suggestion.

Revision: We have rephrased our words throughout the manuscript.

Reviewer #3 (Remarks to the Author):

Re: Charge transfer mediated triplet energy transfer across the inorganic/organic interface, by Luo, et al.

This work proposes a unified model for triplet energy transfer from nanocrystals to molecular acceptors that is mediated by charge transfer states. The experimental system employs CsPbBr₃ perovskite particles with very high quantum yields (and apparently very little spectral shift between their absorption and emission) which is consistent with the absence of surface states. Combinations of the nanocrystals and naphthalene carboxylic acid and tetracene carboxylic acid are studied using transient absorption and PL. The tetracene and naphthalene derivatives are chosen to energetically allow and disallow, respectively, the formation of intermediate charge transfer states.

Overall, the manuscript seems straightforward and a valuable contribution to the field.

The tetracene derivative quenches the nanocrystal PL very rapidly, and the transient and energetic structure appears consistent with a intermediate charge transfer state.

Response: We thank the reviewer very much for these kind comments. Below we provide point-to-point responses to address his/her remaining concerns.

I have some concerns about the NCA experiments. On first examination the energetic structure for the naphthalene derivative interface (Fig. 1c) looks like it could support a charge transfer state via either hole transfer. On closer reading (especially the supplementary), various corrections are proposed that destabilize the hole transfer state. These corrections are not represented in the current figure. I am also concerned about that diagram because it suggests that the oxidation energy of the NCA is the same as the oxidation energy of NCA*, the triplet excited state. It might be helpful to draw a separate diagram for each system with energies of the initial state, CT mediating state, and acceptor triplet.

Response: We thank the reviewer for this very useful suggestion. Indeed, charge transfer driving force calculations are somewhat confusing if we simply look at the “single-particle” energy levels reported in Fig. 1c. We have added the following paragraph in the main text to briefly describe how charge transfer driving forces are calculated and a new Fig. 2 to illustrate the energy of various charge transfer states with respect to that of the initial photoexcited states. The triplet states are not added because of the congestion of such as figure but are emphasize in the text. Please note that the “single-particle” energy levels reported in Fig. 1c have been slightly modified. They are now our experimental values determined from CV measurements, per reviewer 2’s suggestions, instead of values cited from the literature.

Revision:

“The driving forces for CT reactions ($-\Delta G_{CT}$), however, are not simply determined by the “single-particle” energy alignments shown in Fig. 1c. Rather, we need to account for various Coulombic binding and charging energies involved in CT. For example, when examining hole transfer from NCs to NCA, we should consider the energy penalty associated with breaking the electron-hole pair in NCs and putting extra charges into NCs and NCA as well as the energy compensation from electron-hole binding in the charge separated states (NC^-NCA^+). Detailed for these calculations are provided in Supplementary Note 1. In Fig. 2, we plot the calculated energies of various CT states (NC^-NCA^+ , NC^+NCA^- , NC^-TCA^+ and NC^+TCA^-) relative to that of photoexcited NCs (NC^*) in a Marcus-type reaction coordinate diagram; see also Supplementary Table 1. According to the diagram, electron transfer from photoexcited NCs to both NCA and TCA ground states is energetically unfavourable. On the other hand, hole transfer from photoexcited NCs to NCA and TCA ground states should be energetically disallowed and favoured, respectively.

TET from photoexcited NCs to NCA and TCA is energetically allowed in both cases because the triplet energies of NCA (~ 2.6 eV)³ and TCA (~ 1.3 eV)³⁷ are lower than that of the NC band edge exciton (~ 2.7 eV), but may proceed via different mechanisms on the basis of their different CT possibilities.

Figure 2. CT energetics. The parabolas represent the reactant state (NC*-PAH) and the various CT product states drawn on a continuous reaction coordinate mainly contributed by the surrounding medium. The lowest energies of the CT states with respect to that of the NC*-PAH state are indicated.”

I found the emphasis on a 'unified picture' to be unhelpful in the NCA section. It seems to me that in one case, energy transfer proceeds via charge separation, and in the other case, it proceeds by Dexter transfer. The authors propose that the virtual CT is very important in the NCA system, but it is unclear what evidence exists to support this proposal. Indeed, if virtual CT states are so important here, wouldn't they also be crucial intermediates in almost every example of triplet energy transfer that is presently called Dexter transfer?

Response: We thank the reviewer for this insightful comment. Indeed, our results for NC-NCA indicate that, on the spectroscopic aspect, TET occurs directly. That direct TET could also be mediated by virtual CT is simply a speculation based on previously proposed TET models. Since rigorous quantum mechanical calculations are beyond the scope of this work, we probably should not emphasize that we have a unified CT-mediated TET model here, as suggested by the reviewer.

Revision: We have revised the title, the abstract, and the discussion of the manuscript to avoid the use of the term “unified model”. Instead, per the reviewer’s suggestion, we simply state that we have two types of mechanism for TET; one is real CT-mediated TET and the other is direct TET, with the possibility that the direct TET could also be mediated by virtual CT.

This is also manifested using a new Fig. 6:

Figure 6. TET models. (a) If CT is energetically favoured, a real CT state (CT_r) mediates TET from the initial (T_1S_0) to the final (S_0T_1) states. Direct TET from T_1S_0 to S_0T_1 is avoided due to a relatively weak electronic coupling between them. (b) If CT is energetically disallowed, TET can directly proceed from T_1S_0 to S_0T_1 . However, this direct TET could also be mediated a high-energy, virtual CT state (CT_v)."

Arguments about the definition of Dexter transfer aside, I found the arguments about charge separation improving the yield to be far more interesting and possibly worthy of highlighting in the abstract etc....

Response: We thank the reviewer for this useful suggestion. Indeed, in our system where NCs have some hole trapping, direct TET yield is lower than that of hole-transfer mediated TET mainly because the slow TET process cannot compete with ultrafast hole trapping in a sub-ensemble of NCs. However, a downside of the CT-mediated TET mechanism is that this sensitization scheme is often associated with a large energy loss; for example, the energy loss is ~ 1.4 eV for the NC-TCA system whereas it is only ~ 0.1 eV for the NC-NCA system. For these reasons, and also for the limit of the abstract, we didn't emphasize the advantage of the CT-mediated TET in the abstract.

Revision: At the end of the NC-NCA section, we add the following content to discuss this point:

"This yield is much lower than that of NC-TCA (98.6%) mainly because the slow TET process cannot compete with ultrafast hole trapping in a sub-ensemble of NCs. This comparison indicates that in many cases step-wise, CT mediated TET is a relatively more effective strategy for triplet sensitization as it can compete with other charge trapping or recombination pathways using a fast CT step. Note that, however, this sensitization scheme is often associated with a large energy loss; for example, the

energy loss is ~ 1.4 eV for the NC-TCA system whereas it is only ~ 0.1 eV for the NC-NCA system.”

Finally, it might be helpful to plot the absorption and emission of the CsPbBr₃ particles on top of one another. It would be good to confirm that the spectral shift is very small.

Response: We thank the reviewer for the suggestion.

Revision: We add such a plot as a new Fig. 2a:

We also briefly comment the absorption and PL spectra:

“The lowest energy absorption peak of CsPbBr₃ NCs is situated at ~ 460 nm, which is blue-shifted from CsPbBr₃ bulk (~ 520 nm) due to the quantum confinement effect. The PL peak is ~ 474 nm, corresponding to a Stokes shift of 80 meV. The symmetric PL band and the absence of a low-energy, trap-related emission band are consistent the high PL QY ($\sim 70\%$) of the sample.”

REVIEWERS' COMMENTS:

Reviewer #1 (Remarks to the Author):

The authors have sufficiently addressed all reviewer comments and questions.

Reviewer #2 (Remarks to the Author):

I would like to start by commending the authors by responding in detail to the points I raised in my last review. Overall, I feel the authors have adequately addressed my concerns regarding the accuracy of the energy levels they have employed in building their model for triplet energy transfer between perovskite nanocrystals (NCs) and acene energy acceptors. I feel that this manuscript is suitable for publication in Nature Communications.

In reading the authors' revised manuscript, I do have some followup points I would like to provide below that stem from the authors' response to comments in my initial review as well as those by Reviewers 1 and 3.

1) In response to comments raised by Reviewer 3, the authors have revised the title of their manuscript. However, I feel the new title is too broad. The authors demonstrate there is no single mechanism for triplet transfer from perovskite nanocrystals to triplet accepting molecules, which seems to go against the title which alludes to a single mechanism. The title also ignores that quantum confinement likely plays a large role in the mechanism. I would suggest a title that is a bit more specific, something like "Dual Mechanisms of Triplet Energy Transfer from Inorganic Quantum Dots to Organic Molecules." A similar adjustment to the first sentence of the abstract to make it clear that the authors are discussing quantum confined inorganic NCs is also warranted.

2) Looking at the emission spectra of TET (Figure S3), the lack of clear vibronic structure suggests the TET is aggregated. See for example spectra in Ref. 38 of the main text. Aggregation would slightly adjust the singlet energy estimated for TET according to the authors' method, but shouldn't have any major impact on their conclusions.

3) I feel it is important that the authors report rates for charge transfer and exciton transfer that have been normalized by the number of acceptor molecules bound to a NC's surface. This will provide information that can be better compared across systems that bind different numbers of acceptor molecules.

4) I am not a huge fan of the parabolas shown in the redone version of Figure 2. The authors do not specify the reorganization for formation of different charge separated states and it is not clear why parabolas describing the NC⁻:NCA⁺ and NC⁻:TCA⁺ states should have identical curvature given the different structures of the acceptors. I would recommend representing this data as a Jablonski plot that just lists the relative state energies for different charge separated states.

5) In their reply to my comment regarding FRET, the authors argue that they can rule it out based on their kinetic data. However, the authors have the ability to compute the overlap integrals between their NC energy donor and each of their molecular acceptors. This information could be used to compute the potential rate of FRET, if it did occur. I suspect this process may be somewhat slow given the large extinction mismatch between the NC donor and molecular acceptor, which if born out by a calculation, would provide another argument as to why FRET can be ruled out as a competing energy transfer mechanism.

Reviewer #3 (Remarks to the Author):

The manuscript has been extensively revised and I now support publication.

REVIEWERS' COMMENTS:

Reviewer #1 (Remarks to the Author):

The authors have sufficiently addressed all reviewer comments and questions.

Response: We thank the reviewer very much for supporting the publication of this manuscript in Nature Communications.

Reviewer #2 (Remarks to the Author):

I would like to start by commending the authors by responding in detail to the points I raised in my last review. Overall, I feel the authors have adequately addressed my concerns regarding the accuracy of the energy levels they have employed in building their model for triplet energy transfer between perovskite nanocrystals (NCs) and acene energy acceptors. I feel that this manuscript is suitable for publication in Nature Communications.

Response: We thank the reviewer very much for his/her kind comments on our revised paper. Below we address his/her remaining concerns.

In reading the authors' revised manuscript, I do have some followup points I would like to provide below that stem from the authors' response to comments in my initial review as well as those by Reviewers 1 and 3.

1) In response to comments raised by Reviewer 3, the authors have revised the title of their manuscript. However, I feel the new title is too broad. The authors demonstrate there is no single mechanism for triplet transfer from perovskite nanocrystals to triplet accepting molecules, which seems to go against the title which alludes to a single mechanism. The title also ignores that quantum confinement likely plays a large role in the mechanism. I would suggest a title that is a bit more specific, something like "Dual Mechanisms of Triplet Energy Transfer from Inorganic Quantum Dots to Organic Molecules." A similar adjustment to the first sentence of the abstract to make it clear that the authors are discussing quantum confined inorganic NCs is also warranted.

Response: We thank the reviewer very much for the suggestion.

Response: Per the reviewer's suggestion, we have changed the title to "Mechanisms of triplet energy transfer across the inorganic nanocrystal/organic molecule interface"

2) Looking at the emission spectra of TET (Figure S3), the lack of clear vibronic structure suggests the TET is aggregated. See for example spectra in Ref. 38 of the main text. Aggregation would slightly adjust the singlet energy estimated for TET according to the authors' method, but shouldn't have any major impact on their conclusions.

Response: We thank the reviewer for the valuable comment. Indeed, that is what we have discussed in the paper. At the end of page 7, we state that “*Interestingly, the spectral features of TCA on NC surfaces are narrower than those of TCA in toluene solution. This suggests TCA molecules likely aggregate in solution, which is expected for extended π -systems and is consistent with previous reports on singlet fission observed for tetracene derivatives in solution.*^{14,38} *In contrast, TCA molecules are well-dispersed in the ligand shell on NC surfaces,*³⁹ *resulting in narrow line-shapes.*” So we do believe that TCA molecules in toluene are aggregated, leading to the broadened absorption and emission features in Fig. S3. But we also find that after grafting on to NC surfaces, TCAs are likely well-dispersed in the ligand shell on NC surfaces.

Revision: Per the reviewer’s suggestion, we have added the following sentence to the end of page 7:

“The aggregation is also implied by the lack of clear vibronic structures on the PL spectrum of TCA in toluene (Supplementary Fig. 3).”

3) I feel it is important that the authors report rates for charge transfer and exciton transfer that have been normalized by the number of acceptor molecules bound to a NC’s surface. This will provide information that can be better compared across systems that bind different numbers of acceptor molecules.

Response: We thank the reviewer for the valuable comment. There are ~60 molecules per NC, and previous reports (*J. Am. Chem. Soc.* **2007**, *129*, 15132-15133) have shown that transfer rate roughly scales linearly with the number of acceptors. So can approximately divide the hole transfer rate in NC-TCA and TET rate in NC-NCA by 60 to obtain the normalized rate.

Revision: The normalized rates (time constants) have been reported in the paper:

“Accounting for this factor, the hole and electron transfer time constants per acceptor is ~0.53 ns and 1.9 ns, respectively.” (page 13)

“The TET time constant per acceptor is ~126 ns.” (page 15)

4) I am not a huge fan of the parabolas shown in the redone version of Figure 2. The authors do not specify the reorganization for formation of different charge separated states and it is not clear why parabolas describing the NC-:NCA⁺ and NC-:TCA⁺ states should have identical curvature given the different structures of the acceptors. I would recommend representing this data as a Jablonski plot that just lists the relative state energies for different charge separated states.

Response: We thank the reviewer for the valuable comment. Indeed, these parabolas are only for qualitative use. We have now specified in the figure caption that the molecular structures and hence reorganization energies should be different for TCA and NCA.

Revision: We add the following sentence to the Fig. 2 caption:

“Note that the curvatures of these parabolas and the horizontal displacements between

them are not for quantitative use, because the reorganization energies for NCA and TCA molecules should be different.”

5) In their reply to my comment regarding FRET, the authors argue that they can rule it out based on their kinetic data. However, the authors have the ability to compute the overlap integrals between their NC energy donor and each of their molecular acceptors. This information could be used to compute the potential rate of FRET, if it did occur. I suspect this process may be somewhat slow given the large extinction mismatch between the NC donor and molecular acceptor, which if born out by a calculation, would provide another argument as to why FRET can be ruled out as a competing energy transfer mechanism.

Response: We thank the reviewer for the valuable comment. Our previous estimation for a very similar system (*Chem. Sci.* **2019**, *10*, 2459-2464) showed a FRET time constant of 0.46 ns, which is much slower than the hole transfer time reported here (8.9 ps).

Revision: We add the following sentence to page 11:

“Specifically, the FRET time constant estimated for a similar system was 0.46 ns, which is much slower than the hole transfer process observed here.⁴³”

Reviewer #3 (Remarks to the Author):

The manuscript has been extensively revised and I now support publication.

Response: We thank the reviewer very much for supporting the publication of this manuscript in Nature Communications.